# Seepage Actions and Their Consequences on the Support Scheme of Deep-Buried Tunnels Constructed in Soft Rock Strata

**Wadslin Frenelus *** , **Hui Peng *** and **Jingyu Zhang**

Department of Hydraulic Engineering, College of Hydraulic and Environmental Engineering, China Three Gorges University, Yichang 443002, China; zhangjingyu@ctgu.edu.cn
* Correspondence: wadslin@ctgu.edu.cn (W.F.); hpeng1976@163.com (H.P.)

**Abstract:** The stability of deep soft rock tunnels under seepage conditions is of particular concern. Aiming at thoroughly discussing seepage actions and their consequences on the support schemes of such structures, the host rocks of the Weilai Tunnel situated in the Guangxi province of China are used as the research subject. Emphasis is placed on adequately examining the seepage conditions, stresses, displacements and plastic zone radii along the surrounding rocks of such tunnels, taking into consideration the Mogi–Coulomb strength criterion and the elastic-plastic theory. Explicitly, this article proposes analytical solutions for stresses, displacements and plastic radii around deep tunnels in soft rocks under seepage conditions by considering the aforesaid criterion and nonlinear elastoplastic approaches. Subsequently, based on the strain-softening model, the coupled actions of seepage and softening on the rocks surrounding the tunnel are studied. In order to investigate the effects of relevant influencing factors on tunnel stability, parametric studies are thoroughly examined. According to the results, it is revealed that the support scheme of deep soft rock tunnels must be of the highest resistance possible to better decrease the plastic zone and the tangential stress along the host rocks. Moreover, throughout the surrounding rocks, the dissemination of pore water pressure is strongly affected by the uneven permeability coefficient under anisotropic seepage states. The combined effects of softening and seepage are very dangerous for the surrounding rocks of deep-buried tunnels. It is also shown that the seepage pressure substantially affects the plastic radii and tunnel displacements. Under high seepage pressure, the surface displacements of the tunnel are excessive, easily exceeding 400 mm. To better guarantee the reasonable longevity of such tunnels, the long-term monitoring of their support structures with reliable remote sensors is strongly recommended.

**Keywords:** deep-buried tunnels; seepage actions; soft rocks; Mogi–Coulomb strength criterion; strain-softening model; analytical models; tunnel safety and stability





## 1. Introduction

In deep underground engineering, seepage actions are frequent, and their consequences are usually notable. The reduction of the performance of the support scheme and therefore the reduction of the longevity of deep-buried structures are among the major effects of seepage actions in tunnel engineering. It is recognized that the long-term effects of seepage actions are the precursors of many failures in deep rock tunnels. This is due to the fact that stresses and displacements in underground tunnels can be increased by the actions of seepage [1]. In such situations, as related by Otsu et al. [2], it is required to reasonably assess the stability of tunnels facing seepage problems. In fact, worse instability events can happen under hydromechanical coupling when water propagates and concentrates in the existing interstices of surrounding rocks of the tunnel [3]. In deep soft rock strata, water seepage effects are of significant concern and remain an open and pertinent topic. This is because hydration is one of the most determining factors of soft rock alteration mechanisms. To be precise, water seepage hastens the deterioration of soft rock constituents such as clay

minerals and argillaceous cement, which are common particles of soft rocks, and place the tunnels in extremely dangerous conditions. In other words, mineral particles contained in soft rocks are easily degenerated by the actions of water. Therefore, it is essential to adequately and accurately discuss the effects of seepage actions on the support structures of deeply buried tunnels in order to better guarantee their long-term safety and stability.

Abundant research results exist on the effects of water seepage on deeply buried tunnels. For instance, in elastic, homogeneous and isotropic rocky media, Nam and Bobet [4] revealed that, at the tunnel face, the extent and distribution of radial deformations can be significantly increased due to water seepage. Combining theoretical and numerical analyses under the conditions of seepage forces, an analytical solution for ground convergence has been proposed by Shin et al. [5] to forecast the stability of deep tunnels where the host rocks are reinforced with grout and rock bolts. But a specific case has not been detailed for the accurate application of this solution. Wang et al. [6] elastoplastically analyzed the surrounding rocks of deep tunnels using twin shear unified strength theory under seepage conditions. Their results show that, for proper tunnel support structure design and stability control, seepage actions are one of the major factors that needs to be adequately taken into account. Based on strain-softening rock masses and the nonlinear Hoek–Brown failure criterion, Fahimifar et al. [7] proposed an elasto-plastic model to analyze the effects of seepage forces in deep-buried tunnels. Although they have presented some examples for the validation of their model, it remains difficult to accurately apply their solution in deep soft rock tunnels built in complex geomechanical and hydrological conditions. Using a bolted tunnel face under seepage flow conditions, a computational method was developed by Perazzelli et al. [8] to study the stability of lined tunnels. Nonetheless, such a calculation method can be better applied to approximate the support schemes of deep rock tunnels. On their sides and considering seepage forces, Jin-feng et al. [9] employed two failure criteria (generalized Hoek–Brown and Mohr–Coulomb) to establish analytical solutions for evaluating the displacements and plastic radii of circular tunnels working in rock masses and exhibiting elastic-plastic and elastic-brittle–plastic behaviors. According to their outcomes, water seepage greatly affects the deformation of rock tunnels. However, it remains difficult to adequately apply such solutions in actual case studies due to their complicated mathematical expressions. By considering the effects of water seepage actions, Yang et al. [10] studied the long-term stability of deep soft rock tunnels mainly lodged in chlorite schist. Under such conditions, they proposed a visco-elastoplastic rheological model that can describe the comportment of the mentioned soft rock and designed an appropriate support structure aiming at durably ensuring the safety and stability of deep tunnels which are mainly bedded in chlorite schist. The analytical model for determining external stresses in tunnel lining has been established by Yan et al. [11] by combining seepage flow and elastic theories. Their results mainly showed that the evolution of the rock permeability coefficient governs the peak permissible drainage flow along the tunnel environments. Using conformal mapping techniques, Chen et al. [12] proposed workable analytical solutions for anisotropic and steady seepage fields for deep grouted and lined tunnels. Nevertheless, since the exact rock types hosting such tunnels are not clearly provided, it is difficult to accurately employ such solutions in a given study case. Taking into consideration the upper bound theorem of the limit analysis, Di et al. [13] studied the stability of tunnels from an analytical solution combined with an extension of Fourier series and the numerical method for the tunnel face seepage field. Such a solution is very interesting for assessing the stability of the tunnel face taking into account the effects of water seepage. Recently, Guo et al. [14] proposed an analytical method to predict the seepage field around twin tunnels by expanding the Schwartz alternating method associated with issues related to multi-connected domains. Their solutions are suited to deep-buried tunnels crossing diverse rock types possessing different mechanical properties. In such situations, groundwater seepage conditions may vary along the surrounding rocks of deep tunnels. As pointed out by Niu et al. [15], the major factors affecting tunnel stability include variability in groundwater conditions. In fact, varying states of groundwater around surrounding rocks can cause anisotropic states

and make tunnel instability more pronounced. In any situation, after tunnelling, the host rocks of deeply buried tunnels must be reinforced with appropriate grouting materials. Typically, grouting improves the rock mechanical properties and acts as an anti-leakage solution [16,17]. Therefore, as related by Ou et al. [18], tunnel stability is influenced by grout quality. The grouting material must be of sufficient quality to effectively reinforce the rocks surrounding the tunnel openings. In complex deep rock conditions, key technologies such as advanced grouting are utilized to treat groundwater seepage [19]. Nonetheless, regardless of the suitability of the grouting techniques employed, groundwater seepage is inevitable in deeply buried tunnels [20], especially tunnels built in complex soft rock strata. Indeed, this can be explained by two main reasons. Firstly, groundwater can seep through the grouting holes [21] and gradually soften the rock surrounding the tunnels. Secondly, with the passage of time, the tunnel supports lose their original rigidity due to ageing.

Since water seepage is a typical engineering concern [22–28], which generally generates durable unwanted actions in deep rock tunnels [29–33], it needs to be thoroughly taken into account in deep rock engineering. Indeed, water seepage can induce seepage pressure which will endanger the support structure of the tunnels. Markedly, such seepage pressures may produce tensile forces which should be supported by primary supports such as the rock bolts and cable bolts [34]. Therefore, for long-term safety and stability reasons, as explained by Yang et al. [35], the lining structure of deeply buried tunnels can be designed taking into account the seepage field existing around the surrounding rocks. However, in-depth discussions on the actions and consequences of water seepage based on actual cases are not yet abundant in the literature. It should be noted that it is always necessary to quickly assess the stability conditions of tunnels in the event of seepage. In this sense, pertinent analytical solutions are promising because they can quickly provide suitable forecasts [36].

In this paper, seepage actions and their consequences in deep-buried tunnels are analytically discussed, using the surrounding rocks of the Weilai Tunnel in the Guangxi province of China as a relevant case study. Research gaps can be substantially filled by the study of this tunnel which presents complex host rocks where common major engineering problems exist. The main objective of this paper is to show how the actions of seepage affect the support structure and the stability of deep-buried tunnels built in soft rock environments. Thereby, this article establishes analytical solutions for stresses, displacements and plastic radii around deep tunnels in soft rocks under seepage conditions by taking into consideration the Mogi–Coulomb strength criterion and nonlinear elastoplastic approaches. The effects of pertinent influencing factors on tunnel stability are thoroughly examined using relevant parametric investigations. This paper can provide an in-depth understanding of seepage effects on deep underground structures and can serve as a very good reference for related research studies.

## 2. Project Overview, Engineering Context and Rock Parameters

Given its complex conditions in terms of geology and hydrology, the Weilai Tunnel situated in the Guangxi province of China is taken as the research subject. It is buried at a depth of 105 m, is part of the Tianxi Expressway project and is connected to the G357 national highway. The Weilai Tunnel has two lanes, the right one having a length of 662 m and the left one having a length of 686 m. The starting and ending stakes of the right and left lanes are respectively indicated by K114 + 422~K115 + 084 and Z4K114 + 424~Z4K115 + 110. The drill-and-blast excavation method was employed with controlled sequences during the construction of the Weilai Tunnel. A location map of this tunnel and its surrounding rock situation is shown in Figure 1. Figure 2 schematically illustrates the main rock types located in the tunnel alignment. The most predominant surrounding rock of this tunnel is argillaceous sandstone, which is broken, according to a revelation made by a relevant geological survey.

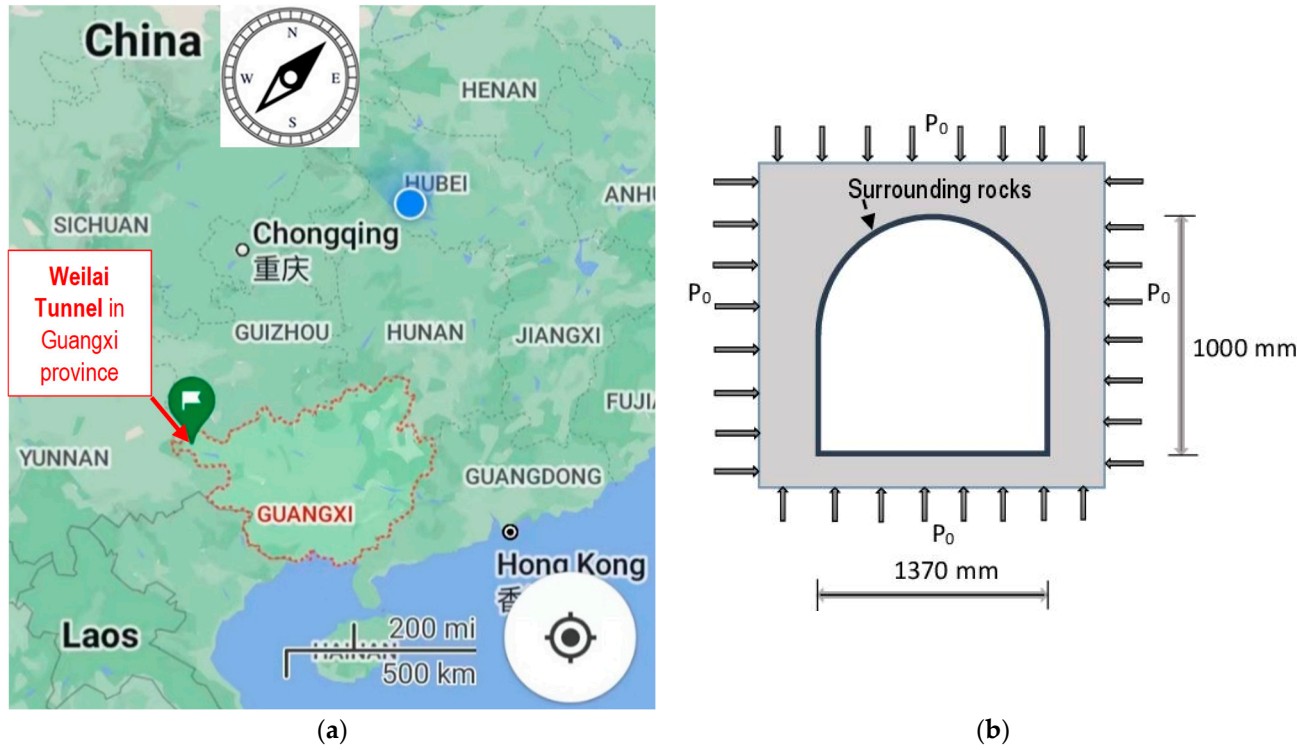

**Figure 1.** Relevant details: (**a**) a view of location map of the Weilai Tunnel; (**b**) excavated section subject to an initial hydrostatic stress field $P_0$.

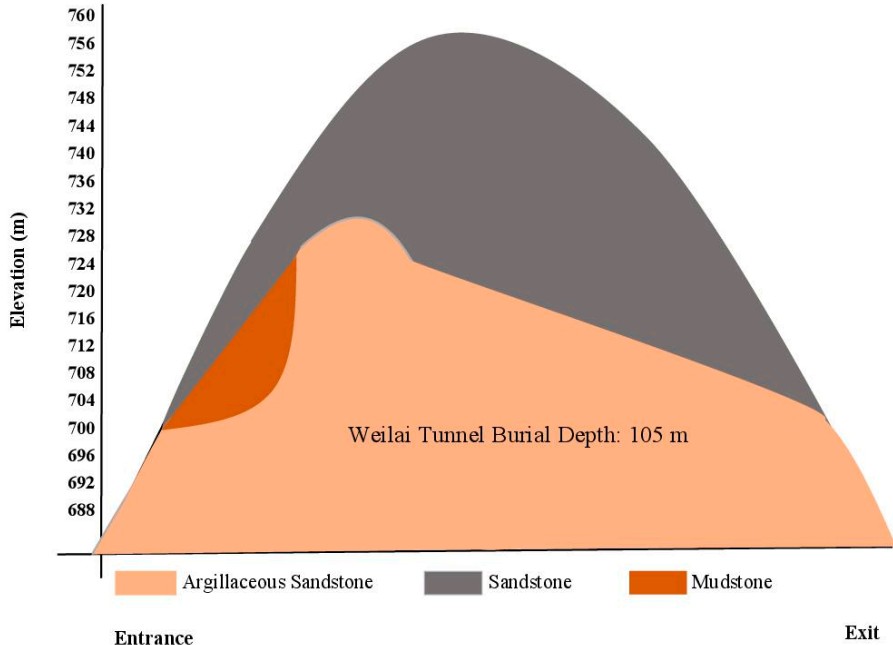

**Figure 2.** Main rock types along the Weilai Tunnel alignment.

Groundwater is relatively rich in the tunnel site due to the existence of aquifers which are close to the tunnel surroundings and the presence of broken rock types. Considerable groundwater ingress was triggered in the excavated region during the construction of this tunnel. Relevant measures have been taken to deal with these groundwater influxes. However, over time, water seepage will inevitably occur in the host rocks [37], and it will be a significant issue in regard to the long-term safety and stability of the Weilai Tunnel.

The results of research related to many rock parameters are valuable. For the surrounding rocks of the studied tunnel, the parameter characteristics in dry and wet conditions are displayed in Tables 1 and 2, referring to Frenelus and Peng [38]. Note that, in wet conditions, most rock parameter values are diminished. Additional parameters of the surrounding rocks and of the tunnel radius are presented in Table 3, based on relevant engineering reports.

**Table 1.** Average basic values of the parameter characteristics of argillaceous sandstone in dry states around the tunnel.

| Rock Type | Uniaxial Compressive Strength (MPa) | Elastic Modulus (GPa) | Poisson's Ratio | Cohesion (MPa) | Internal Friction Angle (°) | Density (g/cm³) |
|---|---|---|---|---|---|---|
| Argillaceous sandstone | 10 | 2.2 | 0.23 | 5.06 | 30 | 0.24 |

**Table 2.** Average basic values of the parameter characteristics of argillaceous sandstone in wet states.

| Rock Type | Uniaxial Compressive Strength (MPa) | Elastic Modulus (GPa) | Poisson's Ratio | Cohesion (MPa) | Internal Friction Angle (°) | Density (g/cm³) |
|---|---|---|---|---|---|---|
| Argillaceous sandstone | 6.6 | 0.62 | 0.39 | 0.93 | 7.5 | 0.38 |

**Table 3.** Additional parameter characteristics of the tunnel and its host rocks.

| Relevant Parameter Characteristics | Unit | Value |
|---|---|---|
| Initial hydrostatic stress | MPa | 10 |
| Hydraulic pressure | MPa | 5 |
| Residual internal friction angle | degree | 5.3 |
| Residual cohesion | kPa | 0.61 |
| Plastic softening parameter | - | 0.005 |
| Tunnel excavation radius | m | 6 |
| Tunnel net radius | m | 5 |

## 3. Adopted Mechanical Model of the Tunnel and Seepage Pressure Determination

Two main parts describe the geometry of the actual tunnel cross section, namely, an upper part which is semi-circular and a lower part which is rectangular. For the convenience of computation and analysis, the whole tunnel is treated as a circular section, in accordance with Peng et al. [39]. Circular-shaped sections are typically referenced in the study of deeply buried tunnels. In fact, in terms of simple comparisons, the cross section of horseshoe, U-shaped and circular cross-sections may be different. As stated by Li [40], the inverted volume of circular-shaped tunnels is generally larger than that of horseshoe tunnels. However, depending on the ratio of the extended radius of the lower part to the radius of the upper part, the horseshoe tunnel may behave differently. When this ratio is equal to or close to 1, a circular section can represent the horseshoe section well [41,42]. Likewise, referring to Bonini et al. [43], the equal-area method is widely used to convert a U-shaped cross section to its equivalent circular cross section. In this paper, the tunnel cross section is originally U-shaped. Based on the aforementioned equal-area method, the U-shaped cross section is converted to its equivalent circular cross section. Accordingly, the equivalent radius ($R_e$) of the circular section which depends on the cross-section area (S) of the U-shaped section, is taken into account. The equivalent tunnel radius is estimated as follows [38,43,44]:

$$R_e = \left(\frac{S}{\pi}\right)^{1/2} \tag{1}$$

Figure 3 displays the adopted mechanical model of the Weilai Tunnel.

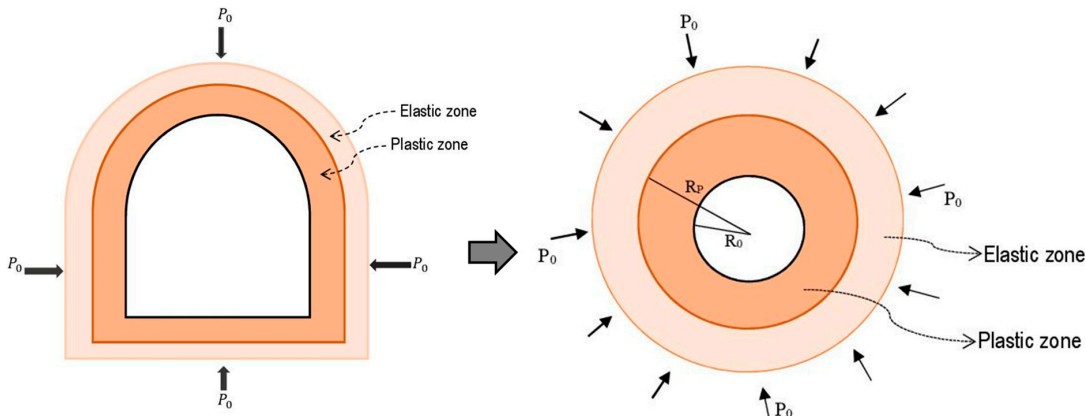

**Figure 3.** Adopted mechanical model of the Weilai Tunnel.

An initial hydrostatic pressure $P_0$ is assumed to exist along the host rocks of the tunnel. The initial seepage field that exists around the tunnel induces an external hydraulic pressure $P_h$. $h$ represents the hydraulic head, which is at infinity. $R_1$ denotes the radius of the tunnel just after excavation, $R_0$ the effective radius of the tunnel after the placement of the secondary lining, while $R_P$ represents the distance between the tunnel center and the boundary of the plastic zone. Subsequently, $R_i$ is assumed to be the distance between the tunnel center and the infinity. The adopted mechanical model of the studied tunnel under seepage states is presented in Figure 4.

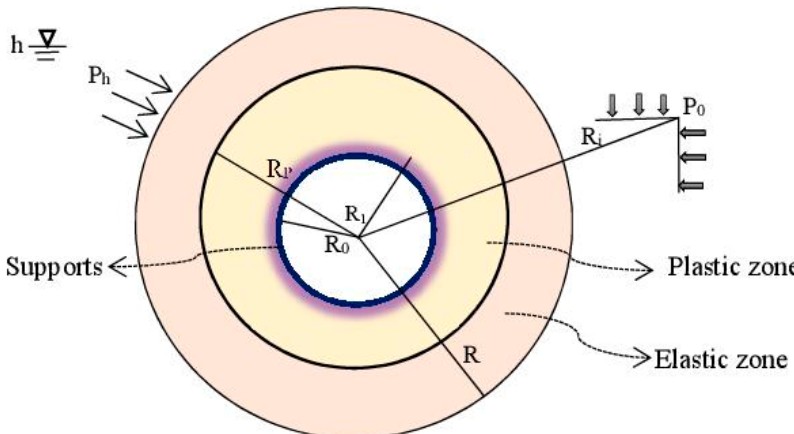

**Figure 4.** Mechanical model of the supported tunnel under seepage states.

Indisputably, due to the external pressure of water, a seepage field exists and will disturb support structure performance. As the tunnel is lined, a stable symmetric seepage field is taken into consideration. Moreover, the permeability of the lining structure and that of the surrounding rocks are also considered. Indeed, referring to Tunç and Tunç [45], the differential equations of the seepage pressure ($P_w$) can be read as below:

$$\frac{d^2 P_w}{dR_i^2} + \frac{1}{R_i}\frac{dP_w}{dR_i} = 0 \tag{2}$$

The relevant boundary conditions can be written as follows:

$$\begin{cases} P_w = 0; \ at \ R_i = R_0 \\ P_w = P_{w1}; \ at \ R_i = R_1 \\ P_w = P_h; \ at \ R_i = R_P \end{cases} \tag{3}$$

Here, $P_{w1}$ stands for the seepage pressure which exists at the interface between the host rocks and the lining.

The continuity and limit conditions are put forward when solving Equation (2). Hence, the seepage pressure can be found equal to:

$$
\begin{cases}
P_w = \dfrac{P_{w1} \ln\left(\frac{R_i}{R_0}\right)}{ln\left(\frac{R_1}{R_0}\right)} & at \ R_0 \leq R_i \leq R_1 \\[4mm]
P_w = \dfrac{P_h ln\left(\frac{R_i}{R_1}\right) + P_{w1} ln\left(\frac{R_P}{R_i}\right)}{ln\left(\frac{R_P}{R_1}\right)} & at \ R_1 \leq R_i \leq R_P
\end{cases}
\tag{4}
$$

By representing the apparent density of groundwater by $\gamma_w$, and the permeability coefficients of the linings and the surrounding rocks, respectively, by $k_l$ and $k_s$, the seepage rate of the linings ($V_l$) and that of the surrounding rocks ($V_s$) can be evaluated as follows:

$$
\begin{cases}
V_l = -\dfrac{P_w k_l d}{\gamma_w R_i} & at \ R_0 \leq R_i \leq R_1 \\[4mm]
V_s = -\dfrac{P_w k_s d}{\gamma_w R_i} & at \ R_1 \leq R_i \leq R_P
\end{cases}
\tag{5}
$$

At the interface between the host rocks and the linings, it is assumed that water seepage operates at consistent rate. Indeed, the union of Equations (4) and (5) yields:

$$
\begin{cases}
P_w = \dfrac{P_h k_l \ln\left(\frac{R_i}{R_0}\right)}{k_l ln\left(\frac{R_P}{R_1}\right) + k_s ln\left(\frac{R_1}{R_0}\right)} & at \ R_0 \leq R_i \leq R_1 \\[4mm]
P_w = \dfrac{P_h k_l ln\left(\frac{R_i}{R_1}\right)}{k_l ln\left(\frac{R_P}{R_1}\right) + k_s ln\left(\frac{R_1}{R_0}\right)} & at \ R_1 \leq R_i \leq R_P
\end{cases}
\tag{6}
$$

## 4. Stress and Displacement Examination

It is of utmost importance to examine the stresses and displacements of the host rocks confronting the actions of water seepage. Around deep-buried tunnels, the stress field is impacted not only by excavation disturbance, but also by seepage actions. For stability reasons, suitable examination in terms of elasto-plastic behavior is really needed. In this sense, as displayed in Figure 5, the adopted mechanical model describes the interaction existing between the host rocks and the supports.

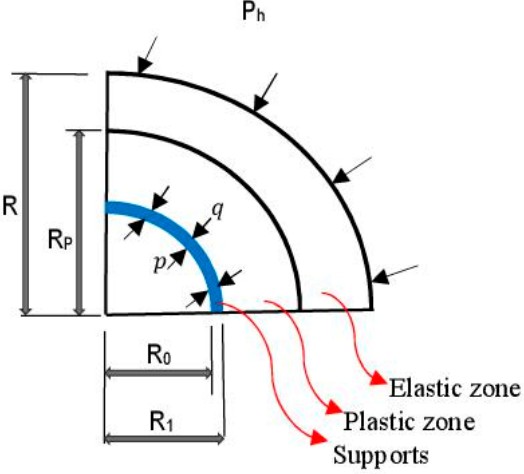

**Figure 5.** Mechanical model of the tunnel for elasto-plastic examination.

The host rocks and the linings will interact continually during the life cycle of the tunnel, as illustrated in Figure 5. Since the states of the stresses and strains are pivotal in the study the stability of tunnels, they have to be adequately examined. To this end,

it is assumed that a pressure $q$ is exerted by the surrounding rocks. On the other hand, a pressure $p$ is thus exerted by the support structures, in terms of reaction. Affected by diverse continuous actions such as seepage pressure, support reactions, and their own pressures, the surrounding rocks of the tunnel will deform plastically with the passage of time.

### 4.1. Examination of Stress in the Plastic Zone

As displayed in the adopted mechanical model, a plastic zone exists along the surrounding rocks of the Weilai Tunnel. The support structures have the ability to reduce the pressure of water seepage around the tunnel. To better take into account such a reduction, a coefficient ($v$) is introduced. Thereby, referring to Zou and Li [46] and Li et al. [47], and considering the reduction of the water seepage pressure, the equation describing the stress equilibrium in the plastic zone can be written as below:

$$\frac{d\sigma_{R_i}}{dR_i} + \frac{\sigma_{R_i} - \sigma_\theta}{R_i} + v\frac{dP_w}{dR_i} = 0 \tag{7}$$

It is worth noting that the plastic conditions are generated by the combined actions of rock supports, water seepage pressure and surrounding rock pressure. Under such conditions, the Mogi–Coulomb strength criterion can be employed [48].

The Mogi–Coulomb strength criterion is widely employed in the study of problems related to underground engineering, particularly in real situations. Its analytical solution takes into consideration the intermediate principal stress coefficient. To be precise, the Mogi–Coulomb strength criterion is a combination of two strength criteria: the Mogi empirical strength criterion and Mohr–Coulomb strength criterion. The Mohr–Coulomb strength criterion is very common. The Mogi empirical strength criterion is expressed as follows:

$$\begin{cases} \sigma_\theta^p = \frac{\sqrt{(\sigma_1-\sigma_2)^2+(\sigma_2-\sigma_3)^2+(\sigma_3-\sigma_1)^2}}{3} \\ \sigma_m = \frac{\sigma_1+\sigma_3}{2} \end{cases} \tag{8}$$

Here $\sigma_\theta^p$ stands for effective radial stress (MPa); $\sigma_m$ is the average principal stress (MPa); $\sigma_1$ and $\sigma_2$ are respectively the maximum principal stress and the intermediate principal stress (MPa); $\sigma_3$ stands for the minimum principal stress (MPa).

The Mogi–Coulomb strength criterion takes into account the parameter characteristics of the rock mass, such as the internal friction angle ($\varphi$) and cohesion ($c$), as well as the effective radial stress ($\sigma_\theta^p$) and the circumferential stress ($\sigma_r^p$). It is expressed as follows:

$$\sigma_\theta^p = A\sigma_r^p + B \tag{9}$$

Parameters A and B are defined as below:

$$\begin{cases} A = \frac{2\sqrt{2(b^2-b+1)}+3k}{\sqrt{2(b^2-b+1)}-3k} \\ B = \frac{6n}{\sqrt{2(b^2-b+1)}-3k} \end{cases} \tag{10}$$

Here $n$ and $k$ are parameters related to Mogi–Coulomb strength criterion [48], and are defined as follows:

$$\begin{cases} A = \frac{2\sqrt{2c}\,\cos\varphi}{3} \\ B = \frac{2\sqrt{2c}\,\sin\varphi}{3} \end{cases} \tag{11}$$

Parameter $b$ describes the coefficient of the principal stress and is defined as follows:

$$b = \frac{\sigma_2 - \sigma_3}{\sigma_1 - \sigma_3} = \frac{\sigma_z - \sigma_r}{\sigma_\theta - \sigma_r} \tag{12}$$

Referring to Li et al. [47] and taking $\nu = 1$, and by placing Equations (6) and (9) into Equation (7), the obtained first-order nonlinear differential equation is solved where two parameters, $A_1$ and $A_2$, are defined as follows:

$$\begin{cases} A_1 = \frac{A_2 B - k_l P_h}{(1-A)A_2} \\ A_2 = k_l ln\left(\frac{R}{R_1}\right) + k_s ln\left(\frac{R_1}{R_0}\right) \end{cases} \tag{13}$$

Accordingly, the plastic stresses exerted on the surrounding rocks can be written as follows:

$$\begin{cases} \sigma_r^p = (p - A_1)\left(\frac{R_i}{R_1}\right)^{A-1} + A_1 \\ \sigma_\theta^p = A\,(p - A_1)\left(\frac{R_i}{R_1}\right)^{A-1} + A_1 A + B \end{cases} \tag{14}$$

Here $p$ stands for the reaction force of the lining to the host rocks.

*4.2. Examination of Stress in the Elastic Zone*

In deep underground engineering, it is widely assumed that the elastic zone can be characterized by a thick-walled cylinder. Consequently, the total elastic radial stress ($\sigma_r^e$), and the total elastic circumferential stress ($\sigma_\theta^e$) exerted by the host rocks can be computed using Kirsch equations [49,50] as follows below:

$$\begin{cases} \sigma_r^e = \left(\frac{R_p^2}{R_i^2}\right)\sigma_{ep} + \left(-\frac{P_0}{2}\right)\left[(1+\lambda)\left(1 - \frac{R_p^2}{R_i^2}\right) + (1-\lambda)\left(1 - \frac{4R_p^2}{R_i^2} + \frac{3R_p^4}{R_i^4}\right)cos2\theta\right] \\ \sigma_\theta^e = -\left(\frac{R_p^2}{R_i^2}\right)\sigma_{ep} + \left(-\frac{P_0}{2}\right)\left[(1+\lambda)\left(1 + \frac{R_p^2}{R_i^2}\right) + (1-\lambda)\left(1 + \frac{3R_p^4}{R_i^4}\right)cos2\theta\right] \end{cases} \tag{15}$$

Here the radial stress of the rocks near the elastic-plastic boundary is designed by $\sigma_{ep}$; the coefficient of lateral pressure is denoted by $\lambda$; the existing angle between the tunnel center line, the vertical direction and the calculated point is represented by $\theta$.

Due to the water pressure and considering that such pressure is uniformly distributed in all directions ($\lambda = 1$), the total elastic radial stress ($\sigma_r^e$), and the total circumferential stress ($\sigma_\theta^e$) generated by the surrounding rocks can be converted to the following equations:

$$\begin{cases} \sigma_r^e = \left(\frac{R_p^2}{R_i^2}\right)\sigma_{ep} - P_0\left(1 - \frac{R_p^2}{R_i^2}\right) + \nu P_w \\ \sigma_\theta^e = -\left(\frac{R_p^2}{R_i^2}\right)\sigma_{ep} - P_0\left(1 + \frac{R_p^2}{R_i^2}\right) + \nu P_w \end{cases} \tag{16}$$

**5. Examining the Plastic Radius and the Displacement of the Host Rocks**

A key factor in deciding on the stability of a deeply buried tunnel is the plastic radius. To do so, it should be reasonably assessed under seepage conditions. The plastic zone must be secured adequately by the support structures in order to guarantee the long-term stability of the tunnel. The extent of the plastic radius is schematically illustrated in Figure 4. It should be noted that the stresses are continuous, and at the elastic-plastic interface, the total plastic stress is equal to the total elastic stress. Hence, the following can be written:

$$\sigma_r^p + \sigma_\theta^p = \sigma_r^e + \sigma_\theta^e \tag{17}$$

Moreover, at the aforesaid interface, $R_i = R_P$. By simply solving Equation (17), the following can be found:

$$[(p - A_1) + A(p - A_1)]\left(\frac{R_P}{R_1}\right)^{A-1} = 2(P_0) - A_1(1 + A) - B \tag{18}$$

Thereby, the radius of the plastic zone is given as below:

$$R_P = R_1 \left( \frac{2P_0 - A_1(A+1) - B}{[p - A_1](A+1)} \right)^{\frac{1}{A-1}} \tag{19}$$

As shown in Equation (19), the radius of the plastic zone relies on several factors. Mainly, it is affected by the reaction force of the tunnel support and the pressure of water seepage.

Around deep-buried tunnels, displacements are inevitable and are categorized into elastic and plastic zones due to the elastic and plastic zones of the surrounding rocks. The plastic radius provides the scope of the plastic deformation around the tunnel. In the elastic zone, the displacements can be estimated by means of elastic theory where the strain can be obtained as below:

$$\begin{cases} \varepsilon_r = \frac{1-\mu^2}{E} \left( \sigma_r - \frac{\mu}{1-\mu} \sigma_\theta \right) \\ \varepsilon_\theta = \frac{1-\mu^2}{E} \left( \sigma_\theta - \frac{\mu}{1-\mu} \sigma_r \right) \end{cases} \tag{20}$$

Here $\varepsilon_r$ and $\varepsilon_\theta$ represent respectively radial and tangential strain; $E$ stands for the rock elastic modulus, $\mu$ represents Poisson's coefficient.

A geometric formula can link strain and displacement as follows:

$$\begin{cases} \varepsilon_\theta = \frac{u}{r} \\ \varepsilon_r = \frac{du}{dr} \end{cases} \tag{21}$$

Hence, in the elastic zone, at $R_1 \leq R_i \leq R_p$, the displacement ($u_e$) of host rocks can be evaluated as follows:

$$u_e = \frac{1-\mu^2}{E} R_i \left( \Delta\sigma_\theta - \frac{\mu}{1-\mu} \Delta\sigma_{R_i} \right) \tag{22}$$

Here $\mu$ stands for Poisson's ratio of the rock mass; $E$ represents the rock elastic modulus.

Likewise, in the elastic zone, at $R_P \leq R_i \leq R$, the host rock displacement ($u_p$) is computed as follows:

$$u_p = \frac{1-\mu^2 R_P^2}{E R_i} R_i \left( \Delta\sigma_\theta - \frac{\mu}{1-\mu} \Delta\sigma_{R_i} \right) \tag{23}$$

## 6. Consideration of the Coupled Effect of Softening and Seepage

As already mentioned, the rocks that surround the tunnel are soft and broken. The softening is amplified by excavation effects such the inevitable complex loading–unloading [51], as well as by water seepage. Consequently, it is more than important to take into consideration the coupling effects of softening and seepage. To this end, it should be noted that elastic and plastic zones are formed around the tunnel after rock excavation, as already stated. Under conditions of softening and seepage, the plastic zone can be appropriately decomposed. Indeed, since rocks typically lose their strength in deep underground engineering [52–56], the plastic residual zone and plastic softening zone are the two main components of the plastic zone. Represented by $R_{pr}$ and $R_{ps}$, respectively, the radius of the plastic residual zone and that of the softening zone, the adopted mechanical model is schematically illustrated in Figure 6.

As presented through Figure 6, the surrounding rocks affected by high stress and water seepage conditions are divided into three zones, namely the plastic residual zone, the plastic softening zone and the elastic zone. Specifically, the elastic zone is formed rapidly in stressed host rocks. Nonetheless, the plastic softening zone describes the conditions of the surrounding rocks in which the stress overtakes the maximum strength of the rock massif. Concerning the plastic residual zone, it is inevitably created because of the progressive deformation of the plastic softening zone. It is worth mentioning that the scope of the plastic deformation along the surrounding rocks of the tunnel is reflected by the plastic radius.

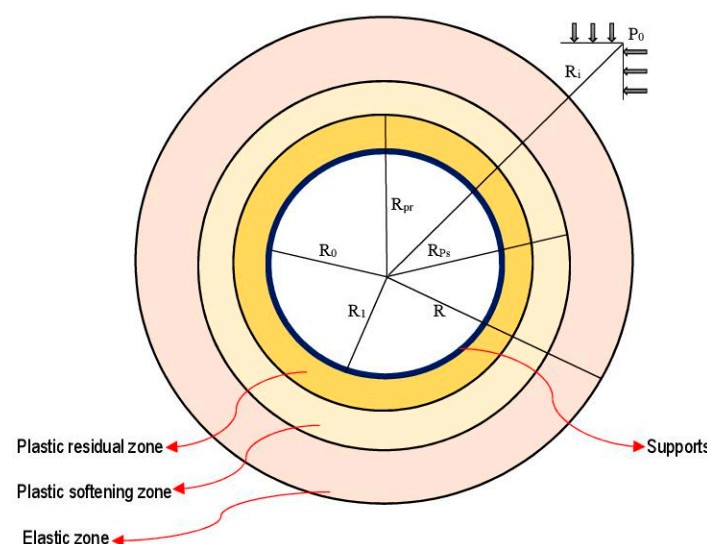

**Figure 6.** Illustration of plastic residual zone and plastic softening zone around the surrounding rocks of deep tunnels.

Rock mechanical properties are affected by softening. However, the rock mechanical properties most affected by softening are the internal friction angle and cohesion. To show how these properties are affected by softening, the strain-softening model is introduced as displayed in Figure 7, where $C_0$ stands for the maximum cohesion, $C_r$ is the residual cohesion, $\varepsilon_\theta^{ep}$ denotes critical tangential strain at the boundary of the elastic zone and plastic softening zone, $\varepsilon_\theta^{pr}$ represents the critical tangential strain at the plastic residual zone/plastic softening zone boundary.

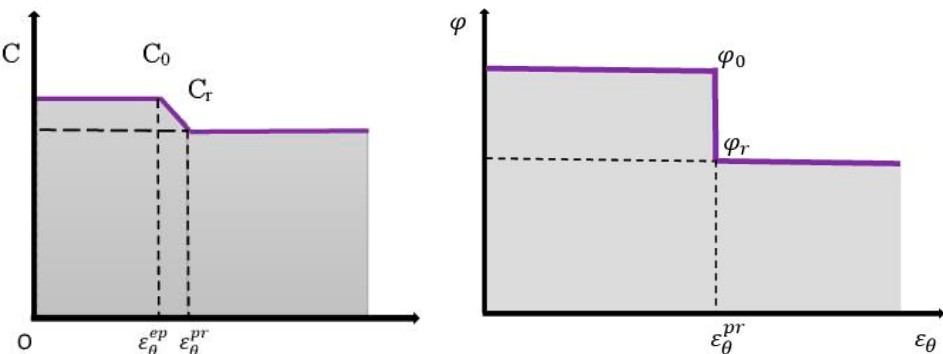

**Figure 7.** Surrounding rocks of the tunnel and the strain-softening model.

According to the model presented in Figure 7, and by taking into consideration relevant linear decreasing functions $C(\eta)$ and $\varphi(\eta)$ for the cohesion and internal friction angle respectively, one can be write [55]:

$$C(\eta) = \begin{cases} C_0, & \eta \leq 0 \\ C_0 + \left(\frac{C_r - C_0}{\eta^p}\right)\eta, & 0 \leq \eta < \eta^p \\ C_r, & \eta \geq \eta^p \end{cases} \tag{24}$$

$$\varphi(\eta) = \begin{cases} \varphi_0, & \eta \leq 0 \\ \varphi_0 + \left(\frac{\varphi_r - \varphi_0}{\eta^p}\right)\eta, & 0 \leq \eta < \eta^p \\ \varphi_r, & \eta \geq \eta^p \end{cases} \tag{25}$$

Here $\eta$ stands for the softening parameter, while $\eta^p$ the parameter reflecting the plastic softening. In this sense, the elastic stage of the tunnel host rocks is reflected at $\eta = 0$.

It is considered that the groundwater flow is incompressible, and the seepage field is stable. The pore water pressure ($P_w$) which depends on the water gravity ($\gamma_w$), and the coefficient of non-uniform permeability ($\beta$) can be expressed as follows:

$$\begin{cases} P_w(R_i) = h\gamma_w \\ \quad \beta = \frac{k_x}{k_y} \end{cases} \tag{26}$$

where $k_x$ and $k_y$ represent the permeability coefficient in horizontal and vertical directions, respectively.

In the plastic residual zone and the softening zone, a continuity equation is put forward to calculate the plane seepage field as below:

$$\frac{\partial V_x}{\partial x} + \frac{\partial V_y}{\partial y} = 0 \tag{27}$$

Here $V_x$ and $V_y$ are, respectively, the seepage flow velocity in the horizontal and vertical directions of the tunnel. Stable seepage flow obeys Darcy's law, and the seepage velocity can be given as below:

$$\begin{cases} V_x = -k_x \frac{\partial h}{\partial x} \\ V_y = -k_y \frac{\partial h}{\partial y} \end{cases} \tag{28}$$

Thereby, the aforesaid velocities can be written as follows:

$$\begin{cases} V_x = -k_x \frac{\partial P_w(R_i)}{\nu} \\ V_y = -k_y \frac{\partial P_w(R_i)}{\nu} \end{cases} \tag{29}$$

Angle ($\alpha$) is taken into account. It is an angle located between the radius $R_i$ and the horizontal axis with coordinates defined as:

$$\xi = \frac{\beta \cos^2 \alpha + \sin^2 \alpha}{\beta \sin^2 \alpha + \cos^2 \alpha} \tag{30}$$

On the basis of Equation (29), the seepage equations can be written as:

$$\frac{\partial^2 P_w(R_i)}{dR_i} + \frac{1}{\xi R_i} \frac{P_w(R_i)}{dR_i} = 0 \tag{31}$$

Then, Equation (31) can be solved taking into account that:

$$\begin{cases} P_w(R_i) = 0 & at & R_i = R_0 \\ P_w(R_i) = P_h & at & R_i = R \end{cases} \tag{32}$$

If isotropic states ($\beta = 1$) are put forward, the pore water pressure can be found as follows:

$$P_w(R_i) = P_h \frac{\ln\left(\frac{R_i}{R_0}\right)}{\ln\left(\frac{R}{R_0}\right)} \quad (R_0 \leq R_i \leq R) \tag{33}$$

Likewise, in the case of non-isotropic states ($\beta \neq 1$), the pore water pressure can be written as:

$$P_w(R_i) = P_h \left[ \frac{R_i^{\left(\frac{\xi-1}{\xi}\right)} - R_0^{\left(\frac{\xi-1}{\xi}\right)}}{R^{\left(\frac{\xi-1}{\xi}\right)} - R_0^{\left(\frac{\xi-1}{\xi}\right)}} \right] \quad (R_0 \leq R_i \leq R) \tag{34}$$

In the plastic residual zone and the plastic softening zone, there is always dilation. Therefore, a non-associated plastic flow rule is taken into account [38]. In this sense, the

radial and tangential strains, respectively, in the plastic softening zone and in the plastic residual zone can be correlated as follows:

$$\begin{cases} \varepsilon_{sr} + k_s\varepsilon_{s\theta} = 0 \\ \varepsilon_{rr} + k_r\varepsilon_{r\theta} = 0 \end{cases} \tag{35}$$

Here, $\varepsilon_{sr}$ and $\varepsilon_{s\theta}$ are, respectively, the radial and tangential strain in the plastic softening zone; $\varepsilon_{rr}$ and $\varepsilon_{r\theta}$ are, respectively, the radial and tangential strain in the plastic residual zone. $k_s$ and $k_r$ stand for the dilation coefficient, respectively, in the plastic softening zone and the plastic residual zone. Denoting $\varphi_s$ and $\varphi_r$, respectively, the dilation angle in the plastic softening zone and the plastic residual zone, the dilation coefficient can be estimated as below:

$$\begin{cases} k_s = \frac{1+\sin\varphi_s}{1-\sin\varphi_s} \\ k_r = \frac{1+\sin\varphi_r}{1-\sin\varphi_r} \end{cases} \tag{36}$$

It is assumed that the dilation angle in the plastic softening zone and in the plastic residual zone are similar.

*6.1. Stress, Plastic Radius and Displacements in the Plastic Residual Zone*

6.1.1. Stress and Plastic Radius in the Plastic Residual Zone

In the plastic residual zone, the coefficient of seepage water pressure ($\alpha$) is taken into account, and the stress balance can be written as below:

$$\frac{d\sigma_{R_i}}{dR_i} + \frac{\sigma_{R_i} - \sigma_\theta}{R_i} + \alpha \frac{dP_w(R_i)}{dR_i} = 0 \tag{37}$$

In general, $\alpha = 1$ for the safety and stability of surrounding rocks. At the interface of the plastic residual zone and plastic softening zone, it should be noted that $R_i = R_{pr}$. Moreover, it is assumed that such an interface is subjected to a maximum principal stress $\sigma_{pr}^{max}$. The plastic tangential residual stress can be written as follows:

$$\sigma_\theta^p = A(P_2 - A_1)\left(\frac{R_{pr}}{R_0}\right)^{A-1} + A_1A + B \tag{38}$$

Here $P_2$ stands for the residual reaction force of the lining to the host rocks.

Also, at the interface between the plastic residual zone and the plastic softening zone, the radial stress is $\sigma_{pr}^r$, and is similar to the tangential stress. The radius of the plastic residual zone can be written as follows:

$$R_{pr} = R_o\left(\frac{\sigma_{pr}^r - A_1A - B}{A(P_2 - A_1)}\right)^{\frac{1}{(A_1 - 1)}} \tag{39}$$

6.1.2. Displacements in the Plastic Residual Zone

In the plastic residual zone, the sum of the plastic softening strain and the plastic residual strain constitutes the global strain, which is assessed as follows:

$$\begin{cases} \varepsilon_{rT} = \varepsilon_{rr} + \varepsilon_r^{sr} \\ \varepsilon_{\theta T} = \varepsilon_{\theta r} + \varepsilon_\theta^{sr} \end{cases} \tag{40}$$

Here, $\varepsilon_r^{sr}$ and $\varepsilon_\theta^{sr}$ are, respectively, the radial and tangential strains at the interface between the plastic softening zone and the plastic residual zone; $\varepsilon_{rT}$ and $\varepsilon_{\theta T}$ are, respectively, total radial and tangential strain in the plastic residual zone.

Based on the non-associated plastic flow rule, the displacement in the plastic residual zone can be estimated as follows [57]:

$$\frac{du_{pr}}{dr} + k_r\frac{u_{pr}}{r} = \varepsilon_r^{sr} + k_r\varepsilon_\theta^{sr} \tag{41}$$

Equation (41) is solved for the plastic residual zone where the following equations are applied:

$$\begin{cases} r = R_{pr} \\ u_{pr} = u_{sc} \end{cases} \tag{42}$$

where $u_{pr}$ is the displacement in the plastic residual zone; $u_{sc}$ is the displacement existing at the interface between the plastic softening zone and the plastic residual zone.

The combination of Equations (41) and (42) yields:

$$u_{pr} = \left[\frac{u_{sr} - \varepsilon_r^{sr} R_{pr}}{k_r + 1}\right] \left(\frac{R_{pr}}{r}\right)^{k_r} + \frac{(\varepsilon_r^{sr} + k_r \varepsilon_\theta^{sr}) r}{k_r + 1} \tag{43}$$

*6.2. Stress, Plastic Radius and Displacements in the Plastic Softening Zone*

6.2.1. Stress and Plastic Radius in the Plastic Softening Zone

Similar to the plastic residual zone, the stress in plastic softening zone conforms to the Equation (37). The maximum tangential stress of the plastic softening zone ($\sigma_\theta^{ps}$) can be written as below:

$$\sigma_\theta^{ps} = A \left(\sigma_{pr}^r - A_1\right) \left(\frac{R_i}{R_{pr}}\right)^{A-1} + A_1 A + B \tag{44}$$

In the limit of the plastic softening zone, where $R_i = R_{ps}$, the tangential stress is equal to the plastic stress $\sigma_p^r$. Hence, the plastic softening zone radius can be written as follows:

$$R_{ps} = R_{pr} \left(\frac{\sigma_p^r - A_1 A - B}{A \left(\sigma_{pr}^1 - A_1\right)}\right)^{\frac{1}{(A_1-1)}} \tag{45}$$

6.2.2. Displacements in the Plastic Softening Zone

In the plastic softening zone, the sum of the elastic strain and the softening strain constitutes the global strain which is assessed as follows:

$$\begin{cases} \varepsilon_{rTs} = \varepsilon_{sr} + \varepsilon_\theta^{ep} \\ \varepsilon_{\theta Ts} = \varepsilon_{rr} + \varepsilon_\theta^{pr} \end{cases} \tag{46}$$

Here, $\varepsilon_r^{sr}$ and $\varepsilon_\theta^{sr}$ are, respectively, the radial and tangential strains at the interface between the plastic softening zone and the plastic residual zone; $\varepsilon_{rTs}$ and $\varepsilon_{\theta Ts}$ are, respectively, the total radial and tangential strain in the plastic softening zone.

The displacement in the plastic softening zone can be estimated, after combining Equations (21), (35) and (46), as follows:

$$\frac{du_{ps}}{dr} + k_s \frac{u_{ps}}{r} = \varepsilon_\theta^{ep} + k_s \varepsilon_\theta^{pr} \tag{47}$$

Equation (47) is solved for the plastic softening zone, where the following equations are applied:

$$\begin{cases} r = R_{ps} \\ u_{ps} = u_{se} \end{cases} \tag{48}$$

where $u_{ps}$ is the displacement in the plastic softening zone; $u_{se}$ is the displacement existing at the interface between the plastic softening zone and elastic zone.

The combination of Equations (47) and (48) yields:

$$u_{ps} = \left[\frac{u_{se} - \varepsilon_\theta^{ep} R_{ps}}{k_s + 1}\right] \left(\frac{R_{ps}}{r}\right)^{k_s} + \frac{(\varepsilon_\theta^{ep} + k_s \varepsilon_\theta^{pr}) r}{k_s + 1} \tag{49}$$

### 6.3. Stress in the Elastic Zone

As already illustrated, a virgin in situ stress $P_0$ is applied to the elastic zone. At the boundary between the elastic zone and the plastic softening zone, the radial stress is represented by $\sigma_r^{ep}$. The stress in the elastic zone can be computed as below:

$$\begin{cases} \sigma_r^E = -\left(\dfrac{R_p^2}{R_i^2}\right)\sigma_r^{ep} + P_0\left(1 + \dfrac{R_p^2}{R_i^2}\right) + \alpha P_w \\[4mm] \sigma_\theta^E = \left(\dfrac{R_p^2}{R_i^2}\right)\sigma_r^{ep} + P_0\left(1 - \dfrac{R_p^2}{R_i^2}\right) + \alpha P_w \end{cases} \tag{50}$$

## 7. Parametric Study and Discussion

The main factors of the analytical solutions were appropriately examined. An adequate parametric study was conducted to show the relevance of the presented analytical approaches. In fact, for proper engineering practices, a parametric study is of great importance. It provides the ability to evaluate diverse key parameters that have a strong influence on the trend and interpretation of analytical or closed-form solutions. Moreover, it also allows us to better understand the established theoretical and analytical models. Thereby, parametric studies constitute solid ways for appropriate and in-depth discussions on the analytical solutions proposed in engineering fields. Hence, it is more than important to carry out parametric investigations of the proposed analytical models. Such parametric investigations are associated with pertinent calculations. The flowchart of the computations is shown in Figure 8. The calculation results are presented in Figures 9–18.

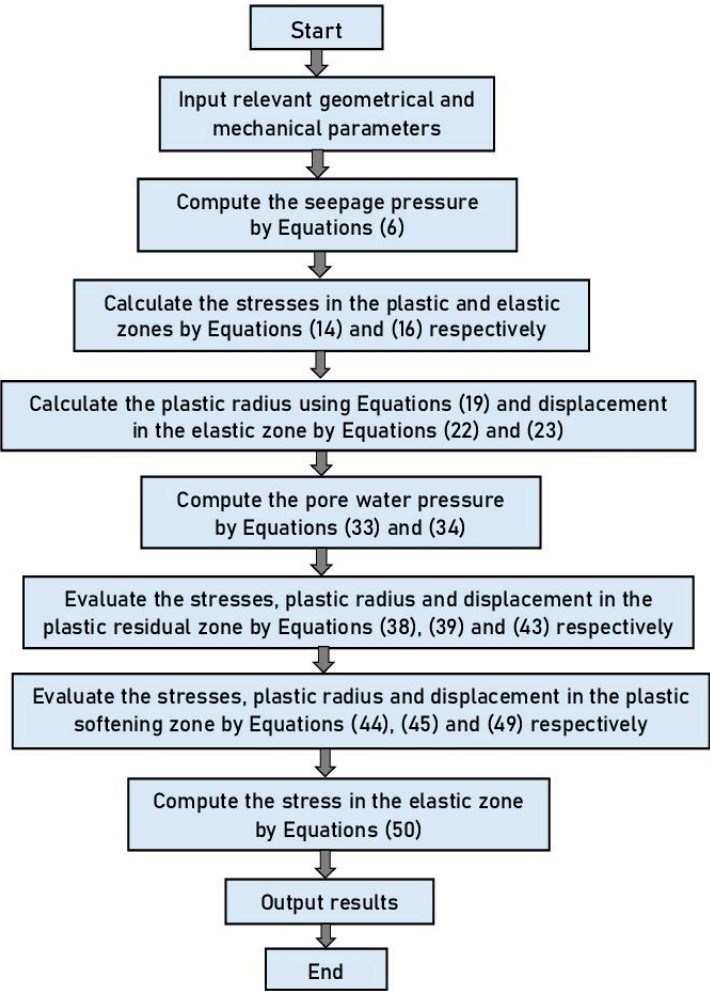

**Figure 8.** Calculation flowchart.

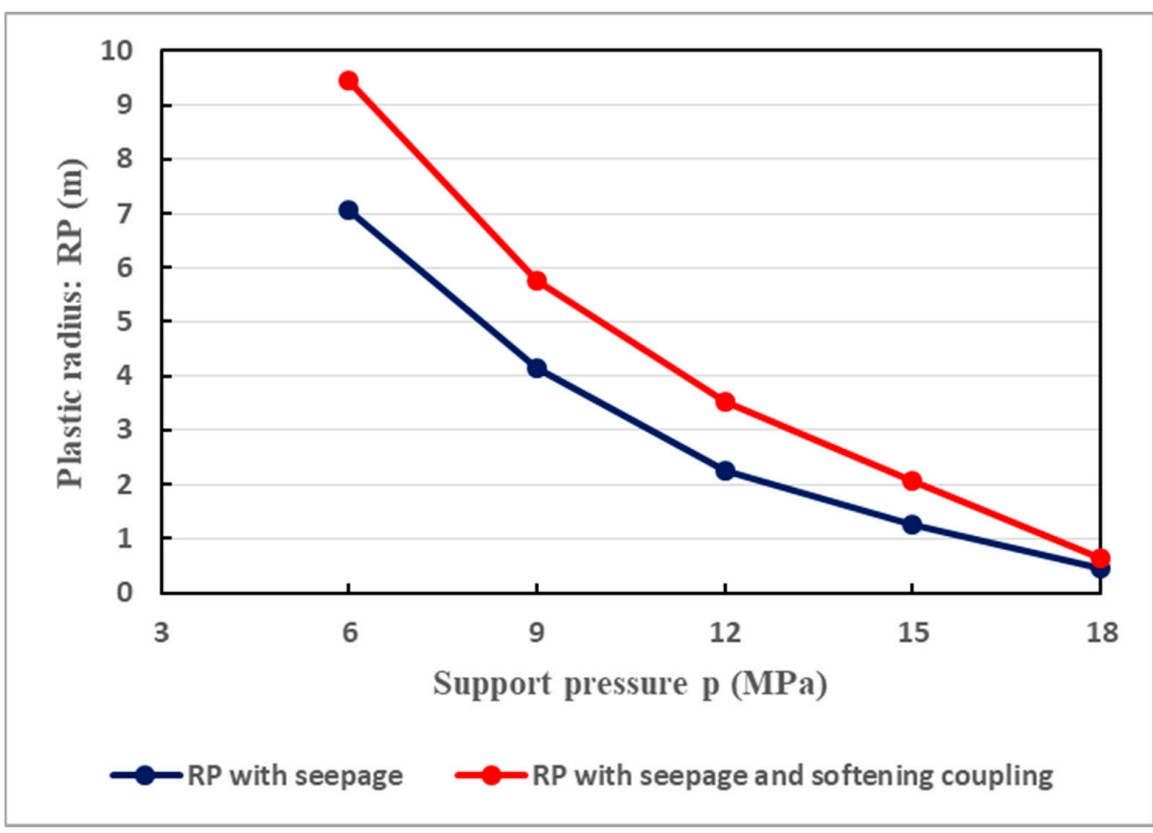

**Figure 9.** Variation of plastic radius ($R_P$) with the support resistance ($p$) when the principal stress coefficient is not accounted for.

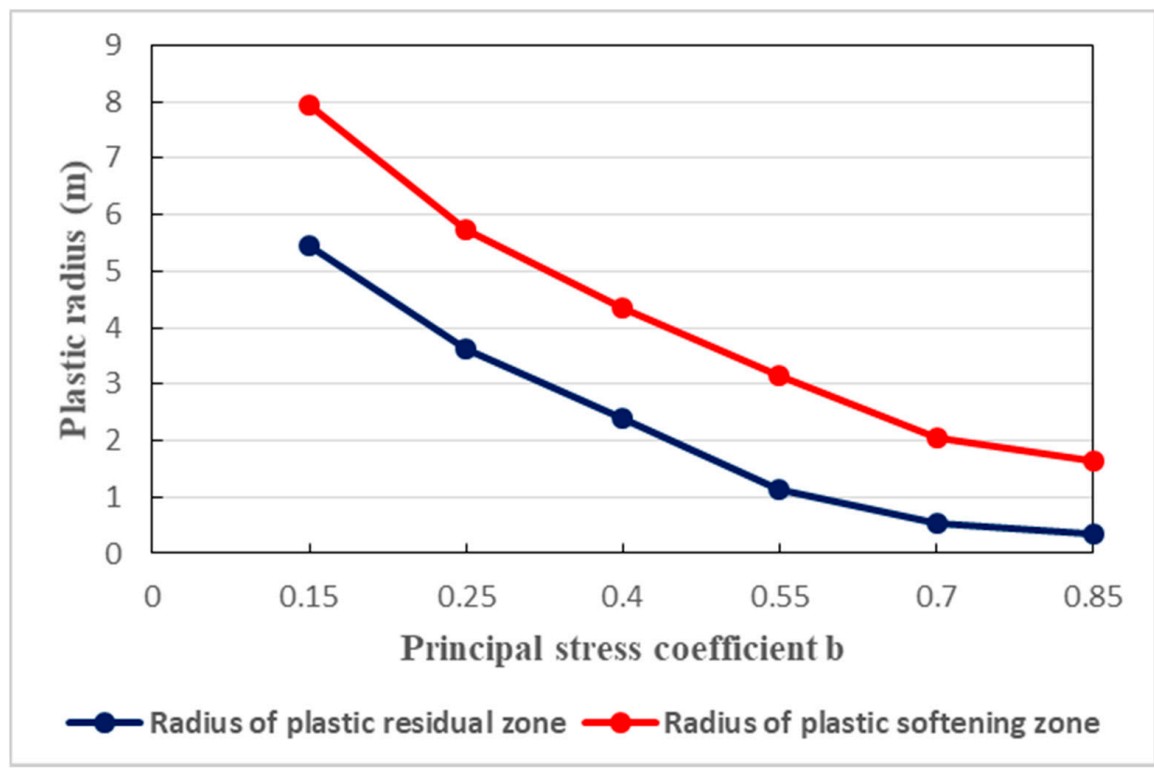

**Figure 10.** Principal stress coefficient b and plastic zone radius.

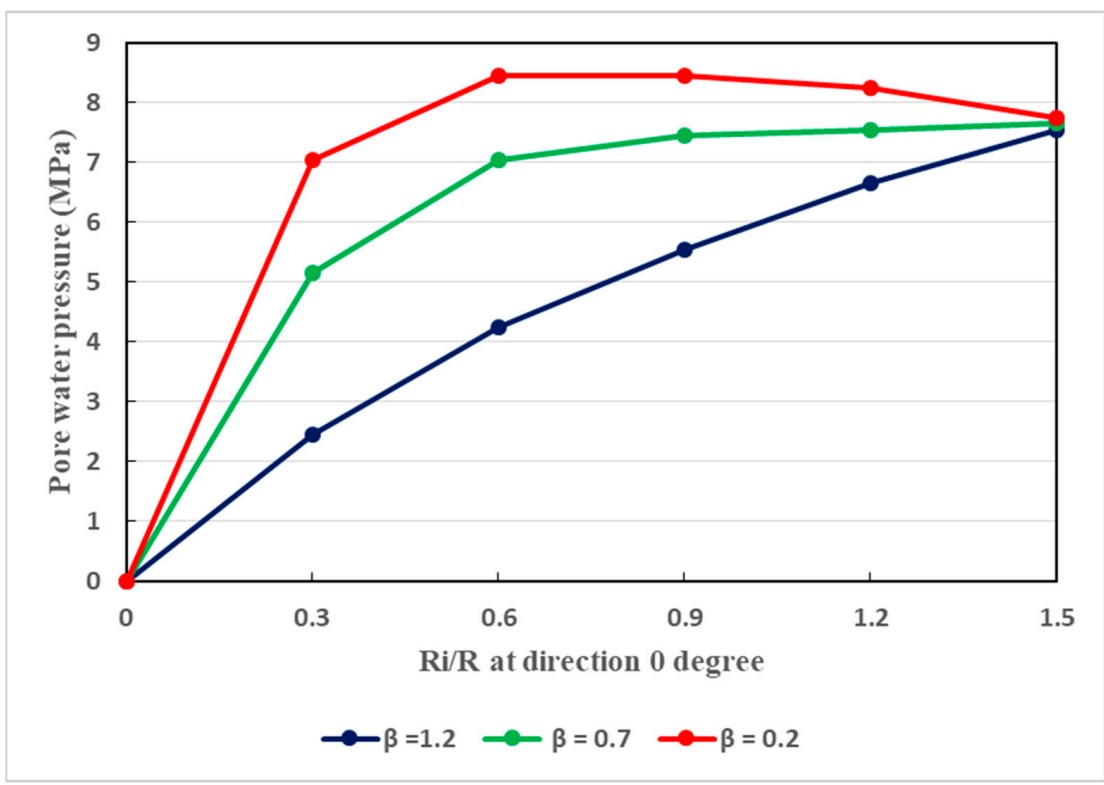

**Figure 11.** Relationship between the pore water pressure and the ratio of plastic radii at different values of the permeability, considering a direction of 0°.

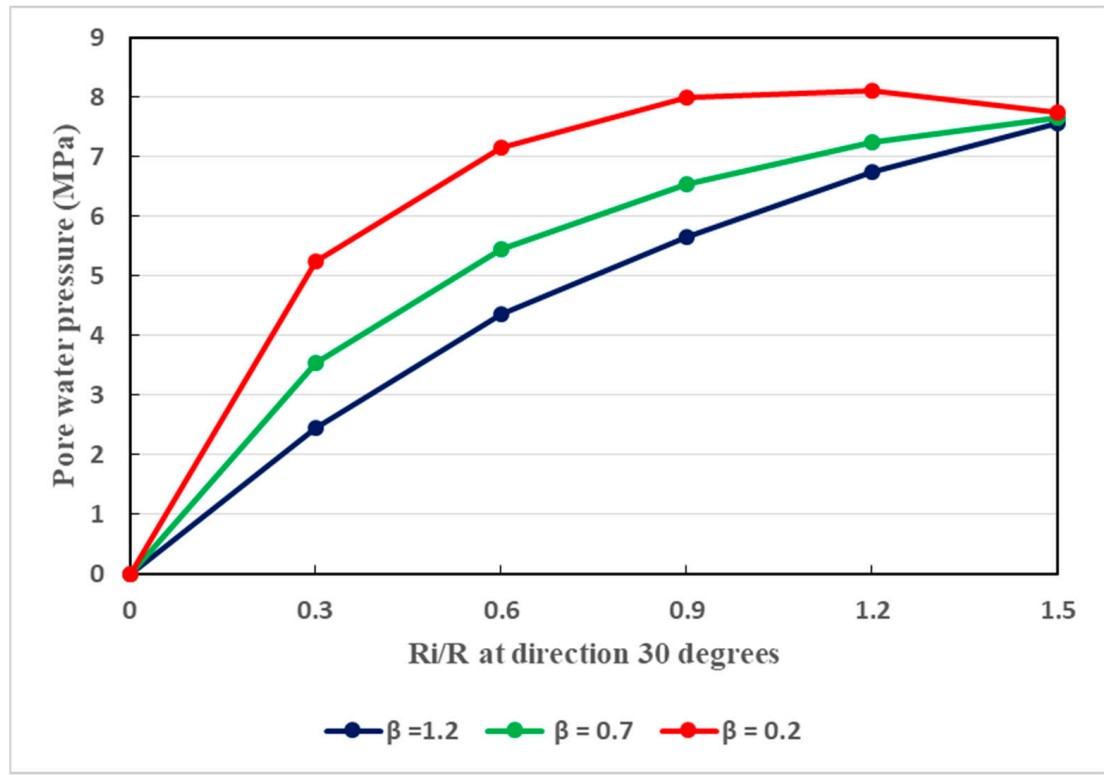

**Figure 12.** Relationship between the pore water pressure and the ratio of plastic radii at different values of the permeability, considering a direction of 30°.

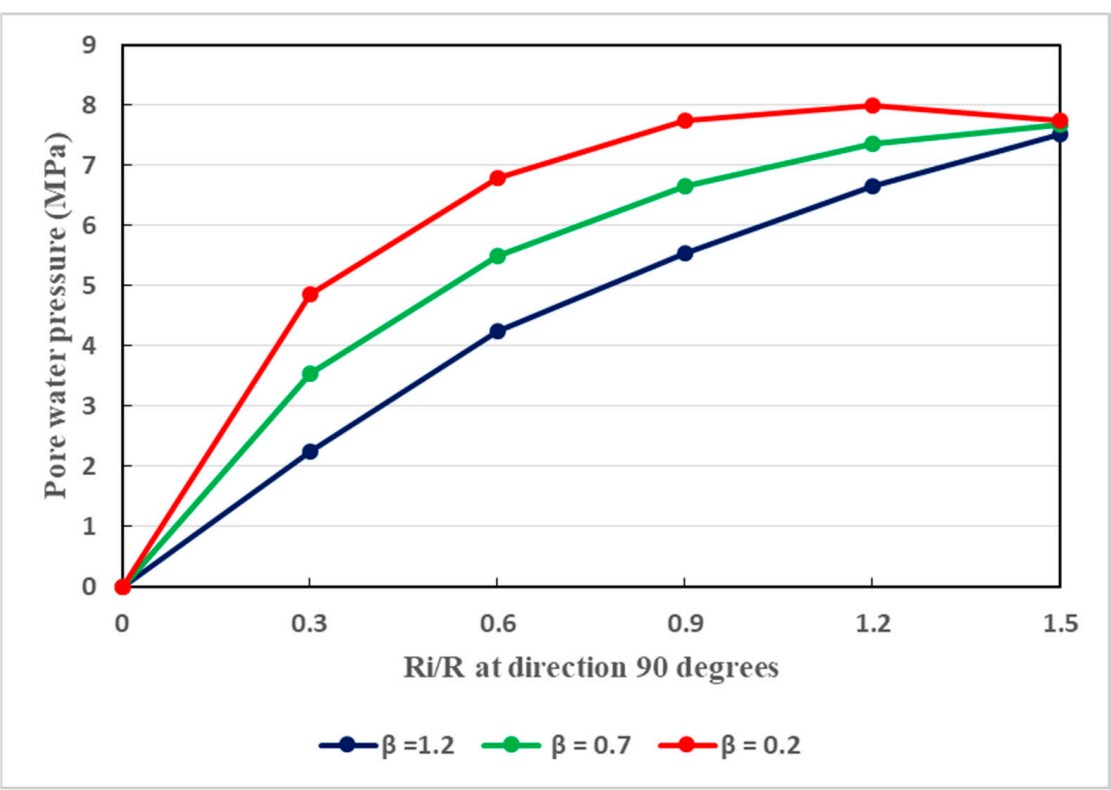

**Figure 13.** Relationship between the pore water pressure and the ratio of plastic radii at different values of the permeability, considering a direction of 90°.

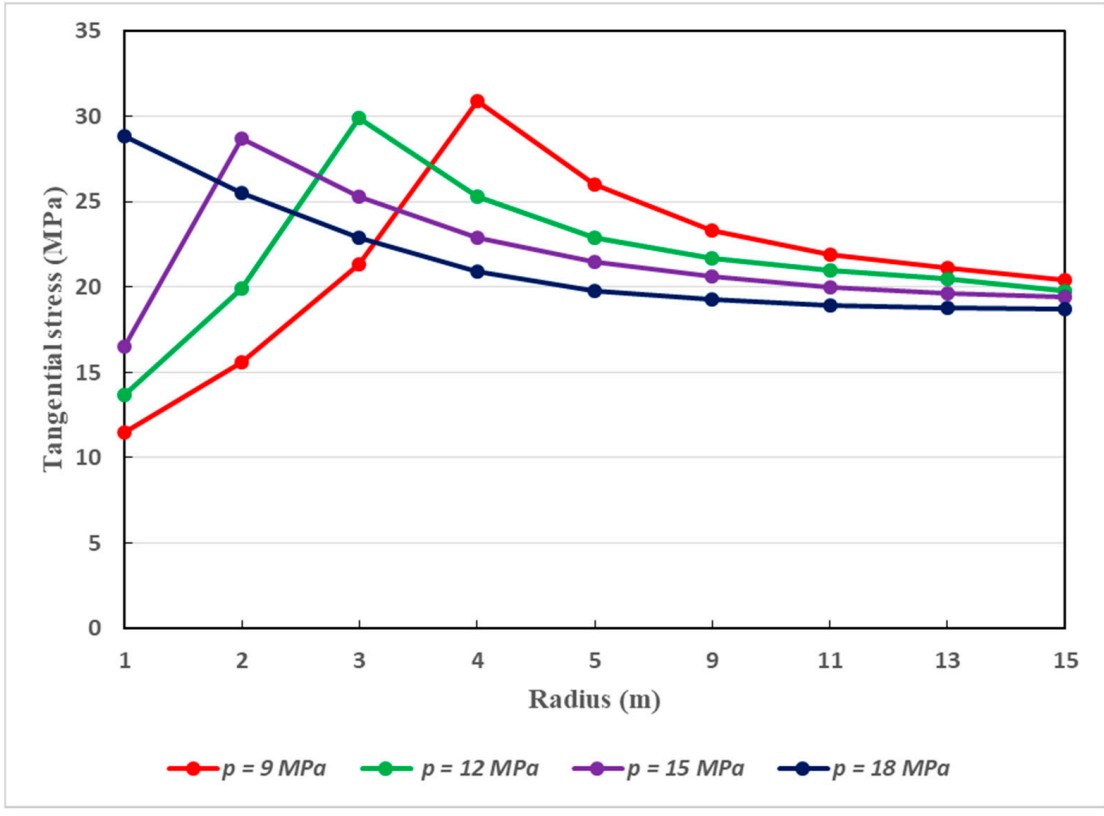

**Figure 14.** Relationship between the tangential stress of the host rocks and the support reaction (*p*) under the coupled actions of softening and seepage.

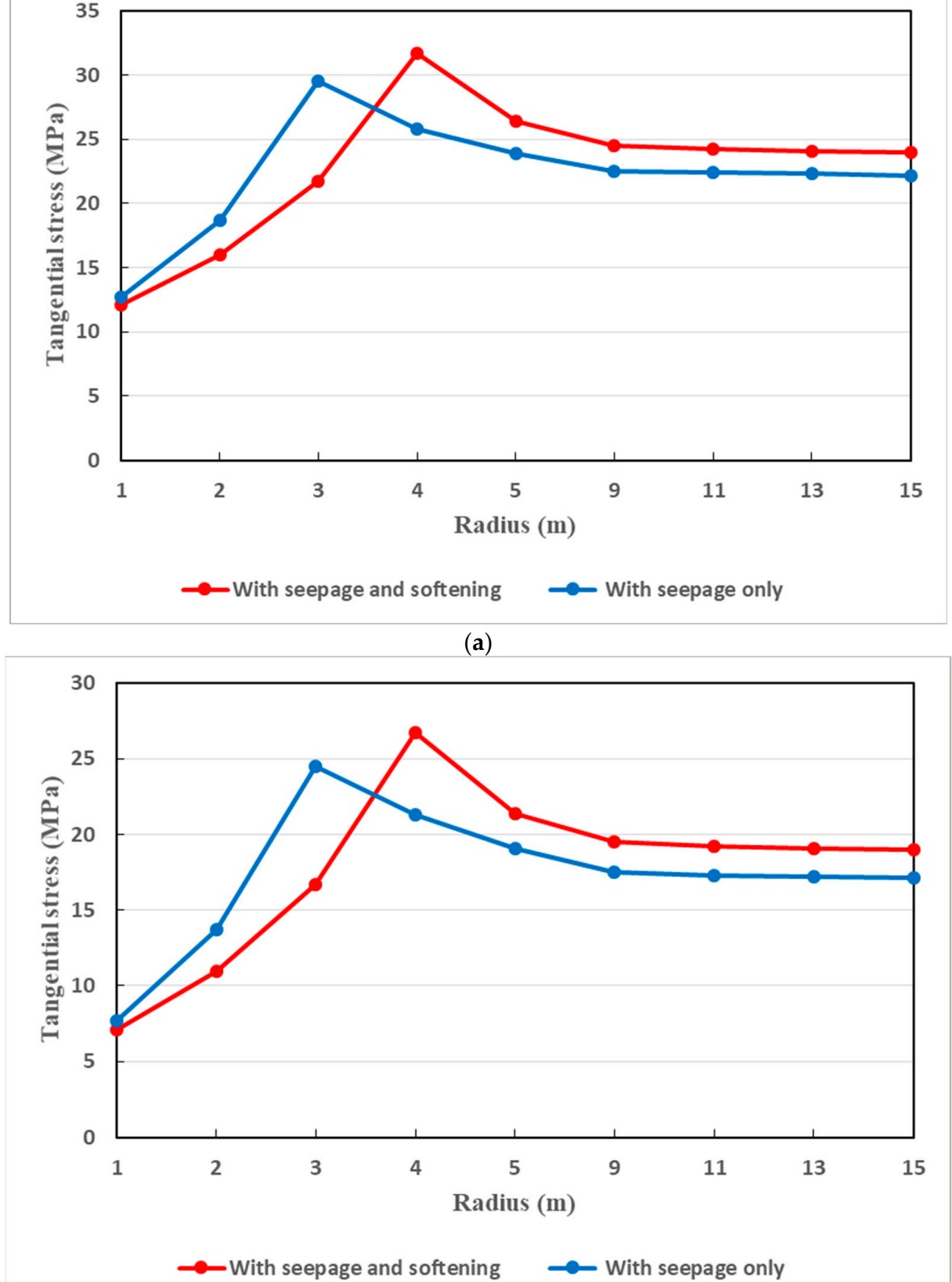

**Figure 15.** (**a**). Dissemination of tangential stress around the surrounding rocks of the tunnel when the support pressure is 9 MPa (principal stress coefficient not accounted for). (**b**). Dissemination of tangential stress around the surrounding rocks of the tunnel when the support pressure is 15 MPa (principal stress coefficient not accounted for).

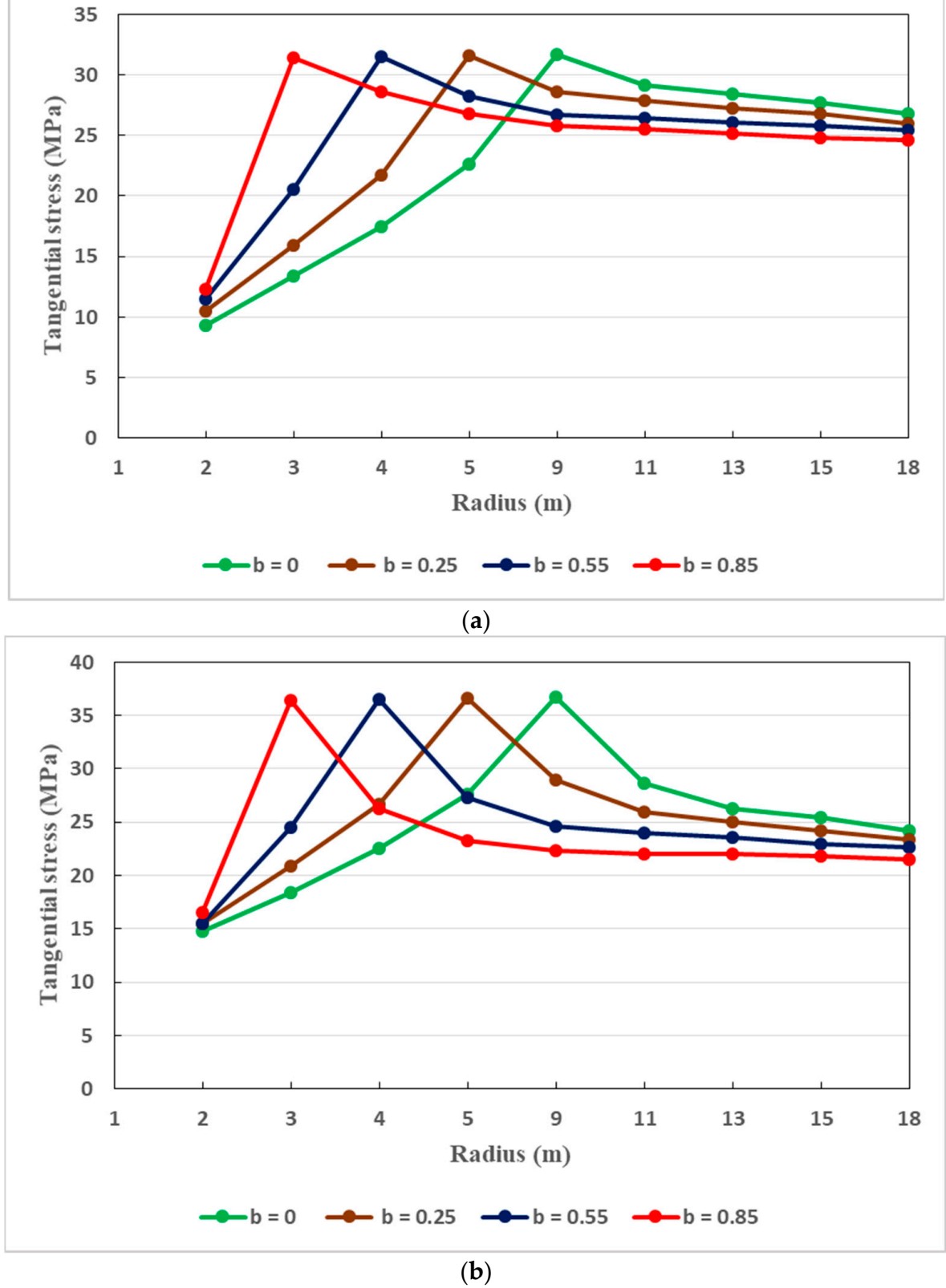

**Figure 16.** (**a**). Dissemination of tangential stress around the surrounding rocks of the tunnel when the support pressure is 9 MPa, for different values of the principal stress coefficient. (**b**). Dissemination of tangential stress around the surrounding rocks of the tunnel when the support pressure is 15 MPa, for different values of the principal stress coefficient.

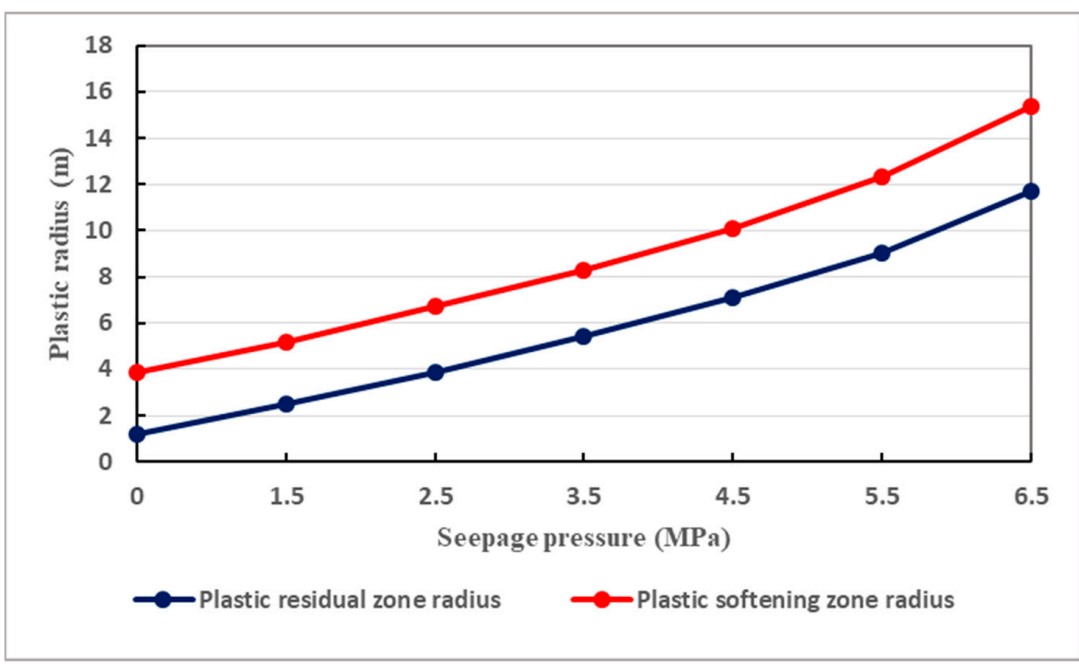

**Figure 17.** Influence of seepage pressure on the tunnel plastic radii.

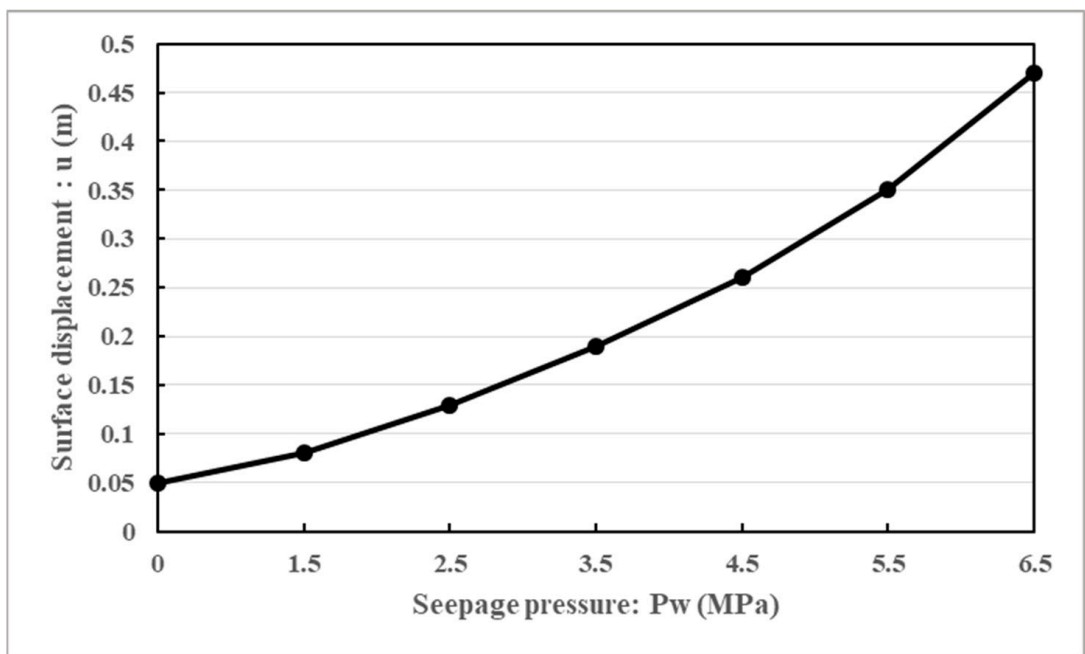

**Figure 18.** Influence of seepage pressure on the tunnel surface displacements.

### 7.1. Variation of Plastic Radius with Support Pressure

Figure 9 illustrates the relationship between the plastic radius and the support resistance (*p*) when the principal stress coefficient is not accounted for. It is shown that the plastic radius undergoes progressive reduction by the augmentation of support resistance. This conforms well to the actual situation. In fact, the safety and stability of the surrounding rocks greatly rely on the resistance of the tunnel support structure. For instance, as it can interpreted in Figure 9, when the resistance of the support structure is 9 MPa, the extent of the plastic radius can reach 4 m. However, when the support resistance is 15 MPa, the scope of the plastic radius is less than 2 m; while when the support resistance is 18 MPa, the extent

of the plastic radius is very low, solely under the action of the seepage. Nonetheless, the coupled action of the seepage and softening can somewhat change this tendency. Indeed, under such conditions, when the support has a resistance of 9 MPa, the plastic radius is significant. A significant plastic radius (2 m) is still observed even when the support has a resistance of 15 MPa. This illustrates the great important of the coupled effects of seepage and softening on tunnel deformation. Thus, the support scheme resistance of the tunnel has to be great enough in order to effectively deal with the coupled actions of seepage and softening.

### 7.2. Variation of the Plastic Radius with the Principal Stress Coefficient

The relationship between the radius of the plastic zone and the principal stress coefficient b is presented in Figure 10. In fact, with an increase in the principal stress coefficient b, a remarkable decrease in the plastic zone radius is observed. For b = 0.15, the radius of the plastic softening zone is found to be approximately 8 m, while it is 3 m when b = 0.55. When b = 0.85, the radius of the plastic softening zone is less than 2 m. Indeed, given the definition of the principal stress coefficient, it is necessary to control the different ratios of the relative stresses in order to limit the extent of the radius of the plastic zone. It is worth noting the plastic zone radius which describes the extent of the deformation around the surrounding rocks, varies with several factors. Reducing such an extent is of particular importance in regard to tunnel stability. To this end, effective countermeasures must be adopted. In deep underground engineering, the most suitable principal stress coefficient is important for the reduction of the plastic zone radius. It should be noted that better control of the plastic zone radius necessitates a sufficient reduction of the plastic zone radius. The lower the plastic radius, the lower the deformation of the tunnel, and therefore the longer the operation of the tunnel can be maintained with safety and stability.

### 7.3. Variation of Pore Water Pressure in the Surrounding Rocks

The host rocks of the tunnel are naturally anisotropic. Aiming at taking this feature into consideration, three different values of the permeability coefficient ($\beta$) are considered, as it is shown in Figures 11–13. It can be interpreted that, in the directions of 0°, 30° and 90°, the tendencies in pore water pressure change are nearly same. Specifically, in all directions, there is gradual increase in pore water pressure as the permeability coefficient increases. However, the evolution rate in the direction of 90° is slightly quicker, as shown in Figure 13. Indeed, Figure 13 illustrates the development of pore water pressure in the direction of 90°. Such a direction describes the vertical direction in which, when the permeability coefficient is augmented, the pore pressure may be developed more quickly than that in the horizontal direction. Globally, in the vertical and horizontal directions, the seepage behavior is not isotropic when the permeability coefficient is changed. Owing to the existence of broken soft rocks in the environment of the tunnel, the permeability coefficient cannot be constant. It can be said that, in the aforesaid directions, the seepage is mainly anisotropic. Its effects are notable and thus cannot be overlooked in regard to the stability of the tunnel. Water seepage effects are very pronounced at great depths. In any direction, water seepage influences the stability of the tunnel. To effectively withstand water seepage actions, a reasonable support structure is required.

It is also observed that the ratios of different radii increase as the water pressure increases. During the rainy periods in Guangxi, water pressure can be increased easily, and can enhance the tunnel plastic zone radii. In such circumstances, the bond between the host rocks and the support system can be deteriorated considerably. This can be more marked in the plastic residual zone and plastic softening zone where the increase in plastic radius is rapid. In fact, in such zones, the rocks are less resistant than those in the elastic zones. Real-time inspection is strongly recommended for such tunnels in order to better control any increase in their plastic radius. Therefore, effective remedial measures at the right time can be ensured.

### 7.4. Variation of Tangential Stress in Surrounding Rocks

Through the tunnel, tangential stresses exist. It is important to study their dissemination along the host rocks of the Weilai Tunnel, as illustrated in Figure 14. In fact, regarding the plastic radius, and considering different values of support pressure, under the coupled actions of seepage and softening, the dissemination of the tangential stress is shown in Figure 14. On the one hand, in the vicinity of the tunnel face, the tangential stress increases rapidly as the support pressure increases. After reaching its maximum value, the tangential stress decreases progressively. This is explained by the fact that, along the host rocks of the tunnel, the support pressures exert a great effect on the distribution of tangential stress. In fact, the latter can be reduced more easily with the highest support pressures. For instance, a support pressure of 18 MPa can be better suited to decreasing the tangential stress along the host rocks of the tunnel. This is another key reason showing that deep-buried tunnels which are constructed in complex soft rocky environments require strong support structure. The supports must be able to generate and maintain reasonable values of the tangential stress around the tunnel for as long as possible, since the longevity of the supports is of particular importance. In fact, it should be noted that the longevity of deep-buried tunnel support structures can be diminished not only by water seepage, but also by other relevant factors such as rock creep [58].

It is also very important to compare the dissemination of tangential stress in two pertinent situations. Figure 15a,b provide such a comparison where the coupled actions of seepage and softening and seepage actions alone are taken into consideration. In the coupled actions of seepage and softening, it can be seen that there is a remarkable decrease in tangential stress. Undoubtedly, this is understandable as the disintegration of rocks is more marked when the coupled actions of seepage and softening are considered.

In fact, the strength of the host rocks is further impaired in such situations. In other words, by the coupled effect of seepage and softening, the tangential stress of the surrounding rocks is reduced when the seepage filed is taken into consideration at the tunnel face. That is, at the tunnel face, the tangential stress under such a coupling effect is less than that with seepage only. But as one moves away from the tunnel face, the tangential stress is somewhat greater under the combination of seepage and softening than under the effect of seepage only. This conforms to the actual situation where the tangential stress of the surrounding rocks is higher under the aforesaid coupling effect.

Figure 16a,b present the variation of tangential stress in the tunnel host rocks with regard to the principal stress coefficient under a support pressure of 9 MPa and 15 MPa, respectively. The tangential stress increases rapidly when the principal stress coefficient (b) increases. After reaching its maximum value, the tangential stress reduces gradually along the surrounding rocks of the tunnel. This can be understood as meaning that the tangential stress is strongly influenced by the principal stress coefficient.

Note that around the host rocks, the tangential stress evolves mainly following two stages under the influence of the principal stress coefficient. In the first stage, it increases quickly as the principal stress coefficient b increases. Then, in the second stage, it becomes stable when the tangential stress attains its maximum value. However, its tendency varies with the value of the principal stress coefficient b. Also, the principal stress coefficient plays a major role in the distribution of the tangential stress along the surrounding rocks. The appropriate value of the principal stress coefficient should therefore be taken into consideration in order to obtain an adequate distribution of the tangential stress around the surrounding rocks of the tunnel.

### 7.5. Influence of Seepage Pressure on the Tunnel Plastic Radii and Surface Displacements

The plastic radii of deep-buried tunnels are greatly affected by seepage pressure. Figure 17 shows how the seepage pressure impacts the plastic radii of the studied tunnel. It clearly explains that an increase in seepage pressure leads to an increase in the plastic radii around the surrounding rocks of the tunnel. The plastic radii are linked to the deformation of the tunnel. Therefore, it can be understood that tunnel deformation also increases with

increasing seepage pressure. Figure 18 also shows that tunnel displacement obviously increases with increasing seepage pressure. Under high seepage pressures, the tunnel surface displacements are enormous. As a result, it is more than important to pay attention to any increase in seepage pressure during the service life of deeply buried tunnels mainly located in complex soft rock strata.

### 7.6. In Situ Measurements of Tunnel Convergence Deformation

It is well known that convergence deformation is an essential factor for judging the stability conditions of deep rock tunnels. To measure the Weilai Tunnel convergence deformation, in situ measurements were carried out. Particular points on the roof and sidewalls were monitored with convergence gauges and total stations [38]. Figure 19 gives illustrative details for the performed in situ measurements.

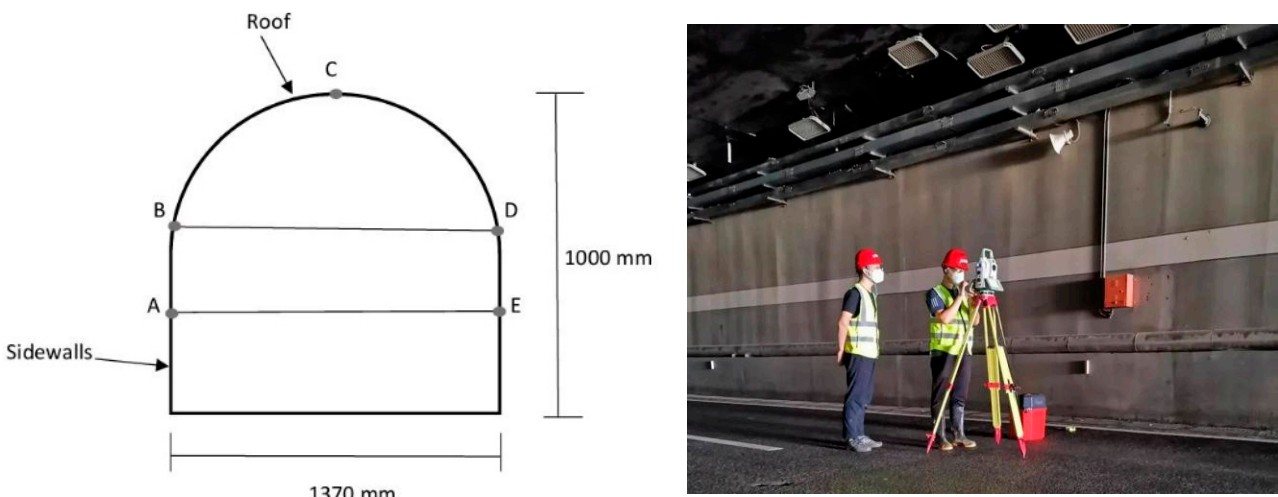

**Figure 19.** Illustrative details of in situ monitoring in the studied tunnel (A, B, C, D and E are critical monitoring points).

Based on in situ monitoring data, the convergence deformation of the Weilai Tunnel is displayed in Figure 20.

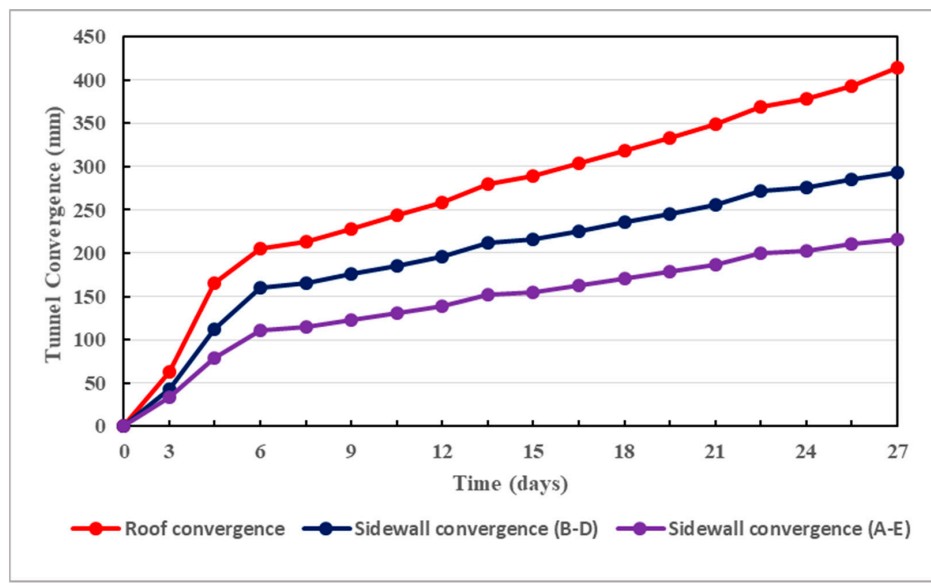

**Figure 20.** In situ measurements of convergence deformation in the Weilai Tunnel.

Under the actual in situ conditions where water seepage exists, the convergence deformation in the Weilai Tunnel is significant. It totaled more than 400 mm in less than a month. This correlates with the results displayed in Figure 18, where the surface displacement of the tunnel exceeds 400 mm under high seepage pressure. Thereby, the seepage actions cannot be overlooked in complex deep soft rock tunnels. They must be carefully taken into consideration in the design of adequate support structures which should ensure the long-term stability of deep-buried tunnels in complex soft rock strata.

### 7.7. Implications

It is important to highlight that seepage actions are inevitable in deep-buried tunnels constructed in complex soft rock strata. Indeed, over time, the support components of these tunnels will experience reduced rigidity [59,60]. Most importantly, seepage can be coupled with stress [61] and can thus provoke more adverse effects on the stability of deep soft rock tunnels. Likewise, the combined effects of seepage and material softening are very detrimental to the tunnel safety and stability. Suitable countermeasures should be adopted to ensure the long-term safety and stability of deep soft rock tunnels facing seepage problems. On the one hand, the support structure must be of highest strength possible to properly perform its function in seepage conditions. For instance, typical primary supports such as rock bolts and cable bolts are generally prone to degradation due to water seepage-induced corrosion. In such situations, their performance will be easily diminished. If adequate measures are not taken at the right time, the primary support will be heavily corroded and the secondary support will become the main support structure of the tunnel, which is very unfavorable. On the other hand, cracks can extend into the secondary support when exposed to seepage. Note that the secondary support material for deeply buried tunnels is generally concrete or reinforced concrete [62], which can also be degraded by the action of water seepage [63]. It should be noted that water seepage is the cause of many structural problems in deep rock engineering [64,65], and is certain to appear over time, even when the most appropriate grouting techniques are applied to the host rocks [27,37]. Hence, at any time, the need to guarantee the safety and stability of deeply buried tunnels is of significant importance. All structural components of such tunnels should be monitored using appropriate remote sensors [66], as well as adequate water/humidity mapping control. The humidity level in deeply buried tunnels must be detected as early as possible in order to prevent its development over time. For instance, when the monitoring system is implemented using water/humidity mapping systems, seepage spots can be detected [67]. Proper decision-making will ensure the reasonable structural integrity and safety of tunnels. In fact, structural health monitoring based on automation and smart techniques is becoming indispensable for the accurate prediction of tunnel safety and stability [68–72]. In addition, the monitoring system must be sufficient to provide all the required information regarding the health status of the tunnel. Thereby, relevant real-time decisions can be made to continuously guarantee the safe operation of deeply buried tunnels. In fact, insufficient monitoring should be avoided because it generally results in unfavorable conditions for the safety of deep soft rock tunnels [73]. In the case of the Weilai Tunnel, which suffers from water seepage problems, comprehensive long-term monitoring carried out by reliable remote sensing techniques is strongly recommended.

This study presents relevant research results in the field of tunnel engineering as it is based on an actual case in China. These results can be exploited worldwide in the study of seepage actions on the support structures of deep tunnels built under similar conditions. These are detailed below:

-    Sufficiently large burial depth of the planned tunnel;
-    Complex soft rock conditions: broken argillaceous sandstone with low uniaxial compressive strength (<10 MPa);
-    Complex rock excavations: drill-and-blast with strictly controlled sequences;
-    Complex hydrological conditions: relatively water-rich zones, groundwater inflows into excavated areas are frequent.

It should be noted that the specific effects of grouting on seepage actions are not examined in detail in this article. This can be assessed in future investigations. In fact, the main motivation of this paper was to study seepage actions and their consequences on the support structures of deep soft rock tunnels. It should be noted that seepage actions are inevitable in deeply buried tunnels constructed in complex soft rock strata, regardless of the suitability of the grouting techniques used. Therefore, the results of this paper can be a source of inspiration for related research studies. Although the stability of deep underground structures can be improved by appropriate grouting [74–76], it should be recognized that in complex soft rock strata with water-rich regions, seepage actions are of pertinent concerns that usually need to be taken into account.

## 8. Conclusions

The stability of deeply buried tunnels constructed in complex argillaceous sandstone media is greatly affected by water seepage actions. In this research study, water seepage actions and their consequences are analytically discussed, mainly based on Mogi–Coulomb strength criterion and elasto-plastic approaches. The main conclusions are as follows:

1. To effectively withstand the inevitable severe consequences caused by seepage actions, the support scheme of deep-buried tunnels, particularly tunnels constructed in soft rock environments, must be of highest resistance possible. It is thus necessary to design such supports in accordance with the complexity of the concerned rocky environments in order to confront the seepage actions.
2. The plastic softening zone and the plastic residual zone are two constituents of the plastic zone of deeply buried tunnels. Their radius decreases by increasing the resistance of the support structure, under seepage conditions and under the combined effects of seepage and softening.
3. The combined effects of water seepage and material softening are very dangerous for tunnel safety and stability. Such effects have the consequences of significantly increasing the plastic radii of deep soft rock tunnels. It is revealed that the more strongly the tunnel is supported, the more its plastic radii and therefore its deformations are reduced.
4. In terms of scope, the plastic zone radii of deep soft rock tunnels are considerably affected by the principal stress coefficient. In fact, the highest values of the principal stress coefficient are favorable to small plastic radii. Accordingly, appropriate values of such coefficients must be adopted to ensure reasonable dissemination of tangential stress along the tunnel, which must be durably safe and stable.
5. Throughout the surrounding rocks of deeply buried tunnels, the dissemination of pore water pressure is strongly affected by the uneven permeability coefficient under anisotropic seepage states. This conforms well to the actual situation where the host rocks are broken and therefore the seepage field is mainly anisotropic. Therefore, seepage actions cannot be ignored since they can affect the stability of deep soft rock tunnels in any direction.
6. Owing to the inevitable severe consequences of seepage actions, deep-buried tunnels constructed in complex soft rocky media must be adequately monitored at all times, even if their support schemes are sufficiently resistant. In fact, proper long-term monitoring can effectively guarantee the safety and stability of such structures at all times. In this sense, reliable remote sensors show promise.

**Author Contributions:** Conceptualization, W.F.; methodology, W.F.; software, W.F.; validation, H.P.; formal analysis, J.Z.; investigation, W.F.; resources, W.F.; data curation, W.F.; writing—original draft preparation, W.F.; writing—review and editing, W.F.; visualization, J.Z.; supervision, H.P.; project administration, H.P.; funding acquisition, H.P. All authors have read and agreed to the published version of the manuscript.

**Funding:** This research was funded by the Special Topics of National Key Research and Development Program of China, grant number 2022YFC3005603-01.

**Data Availability Statement:** Data are contained within the article.

**Conflicts of Interest:** The authors declare no conflicts of interest.

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
