# Peer review of "Seepage Actions and Their Consequences on the Support Scheme of Deep-Buried Tunnels Constructed in Soft Rock Strata"

_infrastructures, doi:10.3390/infrastructures9010013_

Round 1

Reviewer 1 Report

Comments and Suggestions for Authors

In this paper, the authors carried out analytical analysis to investigate the stability of deep soft rock tunnels under seepage application. Generally, this work is interesting and could be considered for acceptance after a few modifications. Some advices is presented as follow,

1. What is Mogi-Coulomb strain criterion? The criterion should be stated in the main text. 

2. Figure 1 is confusing. The red pin on the map should be marked with the project name. 

3. Figure 3 is confusing. The authors should explain why the horeshoe tunnel is modelled by circular tunnel.  In the picture,  it is obvious that the cross-ectional areas of the tunnels are not equal.

4. The surrounding rock is divided into two zones. Do the authors ever concern the grouting effect outside the tunnel lining? The grout layer works significantly on waterproofing. The performance of the grout around the tunnel can be referred to in Ref [1].

5. In the discussion, the in-situ measurements should be added for comparison. Without it, the case study is not that meaningful. 

Reference

[1] Wei Liu, Jiaxin Liang, Tao Xu, Tunnelling-induced ground deformation subjected to the behavior of tail grouting materials, Tunnelling and Underground Space Technology,2023, 140: 105253. https://doi.org/10.1016/j.tust.2023.105253.

Comments on the Quality of English Language

easy to read

Author Response

Dear Reviewer,

Thank you very much for taking your precious time to review my manuscript. Your valuable comments and suggestions have greatly helped me to significantly improve the paper. Please see the attached file for details regarding the revised version.

With my kindest regards

Reviewer 2 Report

Comments and Suggestions for Authors

1. Abstract should be revised to include the originality of study

2. Originality of the study should be presented clearly in the introduction

3. The literature review should include recent articles on tunneling.

https://doi.org/10.3390/su15086797

https://doi.org/10.3390/su141711013

4. Figure 1a should include north-south direction.

5. Figure 1b should include dimension

6. Resolution of Figure 2 should be improved

7. The study focuses in China. Provide explanation how to relate the results of study to the global engineering practice in the world

8. Provide additional section on discussion

Comments on the Quality of English Language

The English requires checking from Professional English checker

Author Response

Dear Reviewer,

Thank you very much for taking your precious time to review my manuscript. Your valuable comments and suggestions have greatly helped me to significantly improve the manuscript. Please see the attached file for details related to the revised manuscript.

With my kindest regards

Reviewer 3 Report

Comments and Suggestions for Authors

I would be a litte more precise in the monitoring techniques and add the water / humidity distribution mapping as a main monitoring system

Author Response

(The authors gave the same response as above.)

Round 2

Reviewer 1 Report

Comments and Suggestions for Authors

This paper is well revised and it is recommended for acceptance.

Reviewer 2 Report

Comments and Suggestions for Authors

The Authors have addressed the comments from Reviewers correctly

Comments on the Quality of English Language

Minor typo and grammatical errors should be fixed